# Fed-DIP: Federated Domain Generalization by Synergizing Implicit Disentanglement and Context-Aware Prompting

## Abstract

Federated Domain Generalization (FedDG) seeks to train a model on decentralized data from multiple source domains that can generalize effectively to unseen target domains. A fundamental challenge lies in achieving robust feature disentanglement—separating domain-invariant from domain-specific features—which is critical for generalization but severely hindered by the data-isolated nature of Federated Learning. Existing methods often struggle with this, leading to incomplete decoupling and limited model performance. To address this, we propose Fed-DIP, a novel framework that introduces an Implicit Decoupling Distillation mechanism. This mechanism achieves fine-grained feature separation by comparing logit outputs for local image regions, all without direct data access. This allows for the robust aggregation of domain-invariant knowledge while critically preserving rich, domain-specific information at the client side. Furthermore, to unlock the potential of this preserved local knowledge, we introduce the Context-Aware Prompt Encoder (CAPE). Unlike prior works that rely on selective prompting from a fixed set, CAPE is a fully generative solution. It dynamically synthesizes adaptive, end-to-end optimizable text prompts directly from local visual features. These generated prompts provide nuanced, contextual guidance, enabling the model to effectively leverage domain-specific insights for more robust and accurate decision-making. Extensive experiments on benchmarks including PACS, VLCS, OfficeHome, and DomainNet demonstrate that our method achieves state-of-the-art (SOTA) performance, validating the effectiveness and superiority of our framework.

## 1 Introduction

Federated Learning (FL), as a distributed machine learning paradigm, enables collaborative model training among multiple clients while keeping their data locally stored (Li et al., 2024a; Luo et al., 2024b). However, the independent and identically distributed (IID) data assumption underpinning traditional FL methods like FedAvg (Luo et al., 2024b) is frequently violated in practice, with client data often exhibiting significant heterogeneity (Non-IID) (Li et al., 2024a; Gong et al., 2024b). Although substantial research has focused on mitigating the performance degradation caused by Non-IID data (Li et al., 2024a; Gong et al., 2024b), these methods still demonstrate insufficient generalization capabilities when confronted with samples from completely unknown or Out-of-Distribution (OOD) data domains.

Domain Generalization (DG) techniques aim to enhance model generalization performance to unseen target domains (Nguyen et al., 2022a; Wang et al., 2022; Sun et al., 2023). A key strategy involves feature disentanglement to separate domain-invariant features from domain-specific features (Li et al., 2023a). However, traditional DG disentanglement methods typically require access to or comparison of raw data across different domains, which conflicts with the data isolation architecture of Federated Learning (FL) (Bercea et al., 2021; Zhou et al., 2023).

Federated Domain Generalization (FedDG) seeks to integrate the distributed architecture of FL with the generalization capabilities of DG, training models across multiple clients that can generalize to unseen target domains (Li et al., 2023b). Existing FedDG approaches, such as sharing frequency

information (Wang et al., 2022) or learning domain-invariant features (Li et al., 2023b; Shangli & Science, 2007), have made progress but face a core challenge: How to achieve effective feature disentanglement under the data-isolated federated architecture? Current methods often lead to incomplete disentanglement, which can introduce statistical noise and potentially discard subtle domain-discriminative information crucial for generalization (Robinet et al., 2024; Bercea et al., 2021). More fundamentally, traditional disentanglement techniques relying on cross-client raw data interaction, such as explicit disentanglement losses or adversarial training, are unsuitable for the FL setting (Bercea et al., 2021; Zhou et al., 2023). Therefore, developing a mechanism that can implicitly promote effective feature disentanglement is a critical unresolved problem in the FedDG field. Moreover, we argue that the choice to share logits is a deliberate privacy-enhancing design. As documented in the literature (Chang et al., 2019; Li et al., 2024c), sharing output logits is inherently more secure than sharing gradients or model parameters. Logits, as high-level abstractions, are more robust against reconstruction and membership inference attacks, thereby reducing information leakage.

Furthermore, while Prompt Learning-based methods show promise in FedDG, their architectural designs critically determine their generalization capabilities. For instance, state-of-the-art approaches like DiPrompT (Bai et al., 2024b) learn a single global prompt for domain-invariant knowledge and a finite pool of domain-specific prompts that act as discrete prototypes for each source domain (Zhao et al., 2023; Blau et al., 2025). An auxiliary query mechanism then *selects* the most appropriate D-Prompt for a given input. This approach is fundamentally selective: its capacity to represent domain context is constrained by the predefined set of learned prototypes, limiting its ability to adapt to nuanced domain variations that lie between or outside these discrete points.

To address this limitation, we propose Fed-DIP, a federated domain generalization framework built around a generative prompting paradigm. Fed-DIP couples two synergistic components: Multi-scale Implicit Decoupling Distillation (MIDD) and a Context-Aware Prompt Encoder (CAPE). Running on the server, MIDD performs feature disentanglement by aligning logit distributions across clients, distilling domain-invariant knowledge into a global adapter while preserving rich, domain-specific signals within each client's local model.

This successful preservation of high-quality local features poses a new challenge: how to effectively leverage this knowledge. Our solution is the client-side CAPE, which functions as a generative, instance-adaptive prompt mechanism. Unlike selective methods that choose from a fixed pool, CAPE is an encoder that dynamically maps the instance-specific visual features preserved by MIDD into a continuous prompt vector. This generative process allows CAPE to represent a continuous spectrum of domain styles, effectively learning a mapping onto what we conceptualize in our theoretical analysis as a continuous prompt manifold. These dynamically generated prompts, combined with globally aggregated features, steer the model towards more refined and robust decisions, ultimately significantly enhancing its generalization ability on unseen target domains.

The main contributions of this paper are as follows:

- We propose the Fed-DIP framework, which synergistically integrates implicit feature disentanglement and generative context-aware prompting to address the core challenges of domain knowledge separation and utilization in federated settings.

- We introduce two novel components: (1) A Multi-scale Implicit Decoupling Distillation (MIDD) mechanism that achieves separation of domain-invariant and domain-specific features without cross-client data exchange. (2) A Context-Aware Prompt Encoder (CAPE), a generative and instance-adaptive mechanism that dynamically maps local visual context onto a continuous prompt manifold, offering finer-grained adaptation than prior selective methods that rely on discrete prompt pools.

- Extensive experiments on four standard FedDG benchmark datasets (OfficeHome, PACS, VLCS, DomainNet) validate the superiority of Fed-DIP, which achieves new state-of-the-art (SOTA) performance.

## 2 RELATED WORK

### 2.1 FEDERATED DOMAIN GENERALIZATION WITH PROMPT LEARNING

The combination of large Vision-Language Models (VLMs) like CLIP (Radford et al., 2021) and parameter-efficient prompt learning (Lester et al., 2021) has advanced Federated Domain Generalization (FedDG). This approach reduces communication by having clients collaboratively train lightweight prompts instead of full models, but faces the challenge of aggregating knowledge from heterogeneous data. A common strategy is decomposing prompts into global and local components to balance generalization and personalization, using methods like specific learning objectives (Cui et al., 2024), Optimal Transport for alignment (Li et al., 2024b), or adaptive prompts (Fang et al., 2025). More advanced architectures use Mixture of Experts (MoE) for personalization (Luo et al., 2024a) or multi-modal visual prompts for richer representations (Singha et al., 2025). A key distinction lies in prompt selection versus generation. While some generative methods use textual domain descriptions (Qiu et al., 2024), our Fed-DIP uniquely synthesizes prompts directly from local visual features. This provides a more fine-grained, instance-adaptive guidance that captures nuanced domain variations, addressing a critical gap in prior work.

### 2.2 PROMPT LEARNING IN VISION-LANGUAGE MODELS

Prompt learning has revolutionized the adaptation of large pre-trained Vision-Language Models (VLMs) like CLIP (Radford et al., 2021). Foundational works can be broadly categorized by how they apply these prompts. One major branch focuses on the language side. CoOp (Zhou et al., 2021) pioneered this by learning continuous prompt vectors to replace hand-crafted text templates. To improve generalization, CoCoOp (Zhou et al., 2022) made these prompts instance-conditional by generating a meta-token from each image. A parallel branch, Visual Prompt Tuning (VPT) (Jia et al., 2022), explores prompting on the vision side by prepending learnable tokens to the input patch sequence of the Vision Transformer (ViT). More recently, hybrid approaches like MaPLe (Khattak et al., 2022) have introduced prompts into both encoders, coupling their learning. Furthermore, ProDA (Lu et al., 2022) explores learning a distribution for each prompt, constructing diverse instances through random sampling to enhance generalization and robustness. Our CAPE module builds on these principles by generating instance-aware prompts conditioned on disentangled visual features, a unique approach in the federated context.

### 2.3 FEATURE DISENTANGLEMENT

Feature disentanglement aims to separate data representations into domain-invariant and domain-specific components (Zhou et al., 2020), often employing techniques like VAEs and GANs (Higgins et al., 2017). However, achieving complete disentanglement is highly challenging. Incomplete disentanglement leads to feature entanglement, impairing model performance (Robinet et al., 2024; Bercea et al., 2021). Moreover, traditional disentanglement methods relying on access to multi-domain data are incompatible with the federated learning architecture.

## 3 METHODOLOGY

### 3.1 PROBLEM SETUP

In the **Federated Domain Generalization (FedDG)** setting, we consider $K$ independent clients. Each client, $C_k$, holds a local source dataset $D_k = \{(x_{ik}, y_{ik})\}_{i=1}^{n_k}$ that is drawn from a unique probability distribution $P_k(X, Y)$, such that $P_k \neq P_j$ for any $k \neq j$. The foundational constraint of this setting is: **Data Isolation**: Raw data sharing among clients is strictly prohibited due to privacy concerns. Clients can only collaborate to train a model by exchanging non-sensitive information, such as model parameters or updates, via a central server. The primary objective in FedDG is to leverage the diverse, isolated source domains $\{D_k\}_{k=1}^{K}$ to learn a single, robust global model $f_{\theta_g}$ that performs well on an entirely unseen target domain $D_t$. This target domain is drawn from a different distribution $P_t$, where $P_t \neq P_k$ for all source clients $k \in \{1, \ldots, K\}$. The global model $f_{\theta_g}$ is produced by aggregating the local models trained on each client's private data. This constrained

optimization problem is formally expressed as minimizing the expected risk on the target domain:

$$\theta_g^* = \min_{\theta_g} \mathbb{E}_{(x,y) \sim P_t} \left[ L(f_{\theta_g}(x), y) \right] \tag{1}$$

where $L$ represents a predefined loss function.

### 3.2 MULTI-SCALE IMPLICIT DECOUPLING DISTILLATION

To enable effective feature disentanglement under federated constraints, we propose the Multi-scale Implicit Decoupling Distillation (MIDD) mechanism. The distillation process is managed by the server and involves bidirectional logit exchange. First, clients compute logits on a public dataset and send them to the server. The server, acting as a student, uses these as teacher signals to update its global model. Then, the server computes updated logits and broadcasts them back to the clients. The clients, now acting as students, use the server's logits as a teacher signal to update their local models. This reciprocal process allows for collaborative refinement.

Let $z$ be the intermediate visual feature from the backbone network. The teacher network is the global model from the previous round, producing logit map $Z_T$. The student network is the current local model, which uses two adapters—a domain-invariant adapter $A_{di}$ and a domain-specific adapter $A_{ds}$—to process $z$.

The MIDD process begins by performing multi-scale average pooling on the logit maps $(Z_T, Z_S)$ to capture fine-grained local semantic knowledge. For each grid cell $i$ at scale $s$, the regional logit values are calculated as:

$$\rho_T(s,i) = \frac{1}{|\mathcal{P}(s,i)|} \sum_{(u,v) \in \mathcal{P}(s,i)} Z_T(u,v) \tag{2}$$

$$\rho_S(s,i) = \frac{1}{|\mathcal{P}(s,i)|} \sum_{(u,v) \in \mathcal{P}(s,i)} Z_S(u,v) \tag{3}$$

Based on these regional logits, we compute a similarity score $\gamma(s,i)^K$ which reflects the agreement between the teacher and student. This score dynamically weights two distinct loss objectives: a consistency loss $D_{cons}$ and a complementarity loss $D_{comp}$. The similarity score $\gamma(s,i)$ is computed as the cosine similarity between the regional logit vectors of the teacher ($\rho_T(s,i)$) and the student ($\rho_S(s,i)$), mapped to a [0, 1] range:

$$\gamma(s,i) = \frac{1}{2} \left( 1 + \frac{\rho_T(s,i) \cdot \rho_S(s,i)}{\|\rho_T(s,i)\| \|\rho_S(s,i)\|} \right) \tag{4}$$

We formally define these losses. Let $f_{di}(z)$ and $f_{ds}(z)$ denote the logits predicted by the domain-invariant and domain-specific branches, respectively. The consistency loss, $\mathcal{D}_{cons}$, uses Kullback-Leibler (KL) divergence to encourage the domain-invariant adapter $\mathcal{A}_{di}$ to learn the consensus knowledge captured by the teacher:

$$\mathcal{D}_{cons} = D_{\mathrm{KL}}(\sigma(Z_T) \| \sigma(f_{di}(z))) \tag{5}$$

where $Z_T$ are the teacher's logits and $\sigma$ is the softmax function. Conversely, the complementarity loss, $\mathcal{D}_{comp}$, encourages the domain-specific adapter $\mathcal{A}_{ds}$ to capture knowledge that is unique to the client and divergent from the global consensus. We achieve this by minimizing the negative KL divergence:

$$\mathcal{D}_{comp} = -D_{\mathrm{KL}}(\sigma(Z_T) \| \sigma(f_{ds}(z))) \tag{6}$$

This complementarity loss is stabilized by two key mechanisms. First, the shared, pre-trained backbone constrains the feature space, making it unlikely for outputs to become orthogonal. Second, the loss weight $\lambda$ is ramped up, ensuring the model first learns stable task-relevant features before the divergence pressure is applied, thus preventing instability during early training.

The total multi-scale distillation loss is then a weighted sum over all regions:

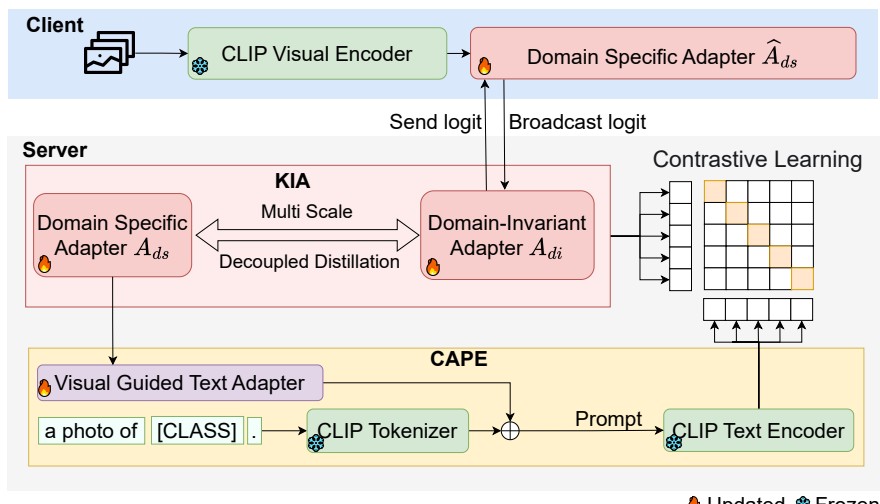

Figure 1: Multi-scale Implicit Decoupling Distillation

$$\mathcal{L}_{\text{MKD}} = \sum_{s,i} \left[ \gamma_{(s,i)}^{K} \mathcal{D}_{\text{cons}} + (1 - \gamma_{(s,i)})^{K} \mathcal{D}_{\text{comp}} \right] \tag{7}$$

Here, the exponent K is a hyperparameter controlling the sharpness of the decoupling. Crucially, this formulation guides the knowledge flow via selective gradient updates: during backpropagation, the gradient from the $\gamma_{(s,i)}^{K} \mathcal{D}_{cons}$ term is used to update only the parameters of $\mathcal{A}_{di}$, while the gradient from the $(1 - \gamma_{(s,i)})^{K} \mathcal{D}_{comp}$ term updates only $\mathcal{A}_{ds}$. This targeted optimization ensures that consensus knowledge is distilled into the global component ($\mathcal{A}_{di}$) and unique knowledge is preserved in the local component ($\mathcal{A}_{ds}$). The final training loss for the client combines this with the standard classification loss:

$$\mathcal{L}_d = \mathcal{L}_{CE} + \lambda \mathcal{L}_{MKD} \tag{8}$$

We ramp up $\lambda$ linearly. A small initial value prioritizes the CE loss for stable training, while a larger final value amplifies the distillation, facilitating the decoupling process and ensuring robust convergence.

### 3.3 CONTEXT-AWARE PROMPT ENCODER

The MIDD mechanism preserves valuable, client-unique knowledge within the local domain-specific adapter $\mathcal{A}_{ds}$. To effectively leverage these insights, we introduce the Context-Aware Prompt Encoder (CAPE), a generative module that transforms this preserved knowledge into adaptive guidance for the model.

CAPE constructs prompts at the embedding level for end-to-end optimization, bypassing text manipulation during the forward pass. At initialization, a set of learnable vectors, the context vectors $\mathbf{c} \in \mathbb{R}^{L \times D}$, are created, where $L$ is the number of context tokens and $D$ is the embedding dimension. These are initialized either from a text string (e.g., "a photo of a") or randomly, and are made trainable. The static parts of the prompt are pre-computed: the embedding for the start-of-sequence token, $\mathbf{e}_{SOS}$, and the embeddings for each class name followed by the end-of-sequence token, $\mathbf{e}_{cls_j,EOS}$.

During the forward pass, the domain-specific visual feature, $f_{ds}$, is processed by a small neural network, the visual adapter $\mathcal{A}_{vis}$, to produce a visual context vector $\mathbf{v}_{ctx} = \mathcal{A}_{vis}(f_{ds})$. This vector is fused with the learnable context vectors: $\mathbf{c}' = \mathbf{c} + \mathbf{v}_{ctx}$. This step injects instance-specific visual information into the prompt. Finally, the complete prompt embedding for each class $j$, $\mathbf{P}_j$, is

assembled by concatenating the components:

$$\mathbf{P}_j = [\mathbf{e}_{SOS}; \mathbf{c}'; \mathbf{e}_{cls_j, EOS}] \tag{9}$$

The resulting tensor $\mathbf{P} \in \mathbb{R}^{N_{cls} \times T \times D}$ (where $T = 77$ for CLIP) is fed directly to the text encoder. This architecture ensures that the prompt length adheres to the model's constraints while allowing the context vectors to be learned through backpropagation, conditioned on visual input.

The input to CAPE is the domain-specific visual feature, $f_{ds}$, which is explicitly defined as the output of the domain-specific adapter operating on the intermediate feature $z$: $f_{ds} = \mathcal{A}_{ds}(z)$. This creates a direct causal link where the knowledge disentangled by MIDD becomes the raw material for prompt generation. CAPE then functions as an encoder, $Encoder_p$, that maps these instance-specific features into a set of optimizable text prompt vectors $U_m^i$:

$$U_m^i = Encoder_p(f_{spec}) \tag{10}$$

These dynamically generated prompts $U_m^i$, along with class description embeddings $D_c$, are fed into the Transformer blocks $\Psi_m$ of the text encoder:

$$\left(s_{m+1}, D_c'\right) = \Psi_m\left(s_m, U_m^i, D_c\right). \tag{11}$$

where $s_m$ is the input summary token. After processing through all $N$ blocks, the final summary token $s_N$ is projected via a text projection layer $\Phi_{text}(\cdot)$ to generate the context-aware class text representation $h_c^i$, infused with instance-specific domain context:

$$h_c^i = \Phi_{text}(s_N, D_{cN}') \tag{12}$$

This process allows Fed-DIP to move beyond fixed templates or discrete domain labels, providing finer-grained, instance-adaptive guidance for model decisions and enabling robust generalization.

### 3.4 OVERALL FRAMEWORK

The client's image encoder consists of $N$ Transformer blocks $\{\Omega_m\}$, incorporating learnable visual prompts $\mathbf{A}_m^i$:

$$\mathbf{s}_{m+1}, \mathbf{E}_{m+1} = \Omega_m(\mathbf{s}_m, \mathbf{A}_m^i, \mathbf{E}_m) \tag{13}$$

After all layers, a projection layer $\Phi_{vision}(\cdot)$ transforms the final summary token $\mathbf{s}_N$ into an intermediate visual feature $\mathbf{z}$:

$$\mathbf{z} = \Phi_{vision}(\mathbf{s}_N) \tag{14}$$

To achieve feature decoupling, we integrate two types of adapters into the backbone network: the Domain-Invariant Adapter ($A_{di}$) and the Domain-Specific Adapter ($A_{ds}$). The Domain-Invariant Adapter captures stable representations universally applicable across all source domains. During each communication round, this global $A_{di}$ is distributed to clients, who fine-tune it using their local data before sending the updates back to the server for aggregation. In contrast, the Domain-Specific Adapter captures unique features specific to each client's domain. Each client locally initializes and maintains its own exclusive $A_{ds}$; this client-specific adapter is never uploaded to the server.

A *Knowledge Integration Adapter (KIA)* fuses these two knowledge streams within the client, computing the final visual feature $\mathbf{z}_v$:

$$\mathbf{z}_v = \mathbf{z} + \frac{1}{2}\mathcal{A}_{ds}(\mathbf{z}) + \frac{1}{2}\mathcal{A}_{di}(\mathbf{z}) \tag{15}$$

Furthermore, we introduce a cross-modal Contrastive Learning framework to ensure that the learned visual features are semantically aligned with their corresponding text representations. This method operates by pulling the visual feature of an image closer to the text feature of its correct class, while simultaneously pushing it away from the text features of all other classes.

Specifically, for a given visual feature $z_{v,i}$, we use the context-aware class text representations $h_c$ (as defined in Section 3.3) as the class prototypes. The contrastive loss $\mathcal{L}_{cl}$ is then defined as:

$$\mathcal{L}_{cl} = -\sum_{i \in I} \log \frac{\exp\left(\text{sim}(z_{vi} h_{y_i})/\tau\right)}{\sum_{c=1}^{C} \exp\left(\text{sim}(z_{vi} h_c)/\tau\right)} \tag{16}$$

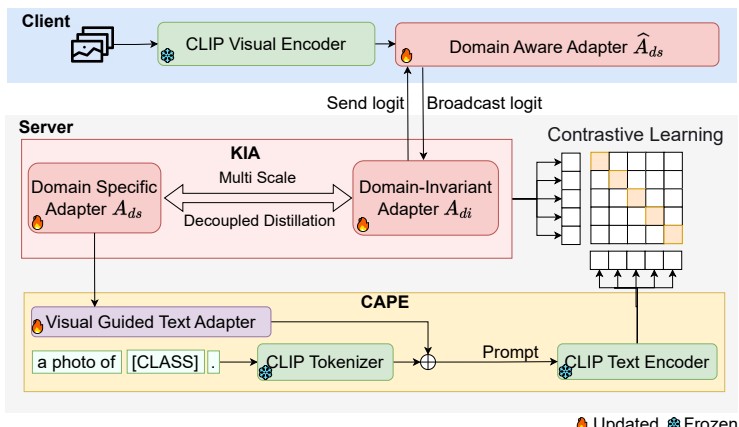

Figure 2: Fed-DIP Architecture Design

where $I$ denotes the index set of all visual features in the current batch, $z_{v,i}$ is the final visual feature for the $i$-th sample, and $h_{y_i}$ is the context-aware text feature for the ground-truth class $y_i$ of that sample. $h_c$ is the text feature for class $c$, and the summation in the denominator is over all $C$ classes. $\text{sim}(\cdot, \cdot)$ is the cosine similarity function, and $\tau$ is a temperature hyperparameter controlling the separation strength.

This loss promotes discriminative representations by encouraging intra-class compactness and inter-class separation in the embedding space, which improves model generalization.

The final total loss $\mathcal{L}$ used for optimizing federated features combines the training loss $\mathcal{L}_d$ and the contrastive loss $\mathcal{L}_{cl}$:

$$\mathcal{L} = \mathcal{L}_d + \beta \mathcal{L}_{cl} \tag{17}$$

## 4 MAIN EXPERIMENTAL RESULTS

In this section, We present our experimental setup and results to validate our approach.

### 4.1 EXPERIMENTAL SETUP

**Datasets.** We evaluate our method on four standard domain generalization benchmarks: **PACS** (Li et al., 2017); **OfficeHome** (Venkateswara et al., 2017); **VLCS** (Fang et al., 2013); and the large-scale **DomainNet** (Peng et al., 2019). For all experiments, we adhere to the standard leave-one-domain-out evaluation protocol, using one domain as the unseen target for testing.

**Baseline Methods.** Fed-DIP's performance is compared against a comprehensive suite of 17 methods, grouped into three categories.

1. **Centralized DG Methods:** These methods, including **SWAD** (Cha et al., 2021), **HCVP** (Zhou et al., 2025), and **Doprompt** (Zheng et al., 2022), operate with centralized data access and serve as a practical upper-bound reference.

2. **CNN-based FedDG Methods:** This group represents foundational federated approaches using CNN backbones. It includes **FedSR** (Nguyen et al., 2022b), **FedADG** (Zhang et al., 2023a), **CCST** (Chen et al., 2023), **ELCFS** (Liu et al., 2021), and **GA** (Zhang et al., 2023b).

3. **Parameter-Efficient Fine-Tuning (PEFT) Methods:** These are the most direct competitors, leveraging parameter-efficient tuning on Vision Transformer (ViT) backbones. This modern cohort includes **FedCLIP** (Lu et al., 2023), **PromptFL** (Guo et al., 2024), **DiPrompT** (Bai et al., 2024a), **FedAPT** (Su et al., 2024), **FedPR** (Feng et al., 2023), **FedMaPLe** (Khattak et al., 2023), and **PLAN** (Gong et al., 2024a).

**Implementation Details.** Our framework is implemented in PyTorch. For fair comparison, we use the pre-trained ViT-B/16 model from CLIP (Radford et al., 2021) as the shared backbone. The fed-

Table 1: Performance comparison on PACS, VLCS, and OfficeHome datasets. The best results are in **bold**. Our method, Fed-DIP, consistently outperforms all baselines across the average of all three datasets.

| Method | PACS | | | | | VLCS | | | | | OfficeHome | | | | |
|---|---|---|---|---|---|---|---|---|---|---|---|---|---|---|---|
| | A | C | P | S | Avg. | C | L | V | S | Avg. | A | C | P | R | Avg. |
| *Centralized DG Methods* | | | | | | | | | | | | | | | |
| SWAD | 93.23 | 85.93 | 99.18 | 82.03 | 90.44 | 98.49 | 68.36 | 75.40 | 79.49 | 79.31 | 76.26 | 68.87 | 86.74 | 87.03 | 79.73 |
| HCVP | 93.17 | 86.89 | 99.33 | 81.30 | 90.17 | 96.32 | 66.26 | 76.40 | 81.65 | 81.08 | 81.77 | 69.76 | 88.01 | 90.62 | 82.54 |
| Doprompt | 95.00 | 86.35 | 99.63 | 78.20 | 89.91 | 96.70 | 66.53 | 78.28 | 79.39 | 80.23 | 80.95 | 70.88 | 88.94 | 90.10 | 82.72 |
| *CNN-based FedDG Methods (Backbone: ResNet-50)* | | | | | | | | | | | | | | | |
| FedSR | 88.19 | 67.45 | 95.74 | 65.92 | 79.33 | 95.16 | 65.86 | 78.51 | 73.49 | 78.26 | 69.12 | 49.69 | 72.71 | 79.12 | 67.66 |
| FedADG | 82.93 | 65.42 | 98.09 | 65.36 | 77.95 | 95.21 | 65.76 | 76.43 | 75.96 | 78.34 | 69.31 | 48.76 | 72.89 | 79.13 | 67.52 |
| CCST | 87.02 | 74.57 | 98.29 | 65.84 | 81.43 | 96.49 | 65.73 | 76.42 | 77.67 | 79.08 | 69.23 | 51.36 | 72.09 | 81.27 | 68.19 |
| ELCFS | 86.77 | 73.21 | 98.14 | 65.16 | 80.82 | 95.67 | 65.02 | 76.55 | 77.96 | 79.80 | 68.17 | 50.52 | 71.44 | 80.11 | 67.56 |
| GA | 87.68 | 75.19 | 97.56 | 65.86 | 81.57 | 96.77 | 65.16 | 78.89 | 78.93 | 79.18 | 68.62 | 50.60 | 73.35 | 81.23 | 68.45 |
| *PEFT-based FedDG Methods (Backbone: ViT-B/16)* | | | | | | | | | | | | | | | |
| FedCLIP | 96.19 | 97.91 | 99.76 | 85.85 | 94.93 | **99.93** | 66.98 | 73.28 | 87.14 | 81.83 | 78.45 | 64.77 | 73.28 | 87.84 | 79.69 |
| PromptFL | 96.34 | 98.46 | 99.58 | 92.19 | 96.64 | 99.71 | 68.03 | 72.24 | 85.10 | 83.59 | 82.98 | 68.98 | 92.14 | 90.27 | 83.59 |
| DiPrompT | 94.97 | 96.25 | 99.56 | 84.72 | 93.88 | 99.70 | 69.23 | 84.16 | 81.72 | 83.70 | 74.21 | 58.90 | 85.51 | 86.12 | 76.18 |
| FedAPT | 97.15 | 99.12 | 99.69 | 92.34 | 97.08 | 99.36 | 68.18 | 81.06 | 85.98 | 83.64 | 83.96 | 71.65 | 91.93 | 90.51 | 84.51 |
| FedPR | 98.10 | 99.02 | 99.88 | 91.11 | 97.03 | 99.36 | 68.18 | 81.06 | 85.98 | 83.64 | 84.04 | 71.63 | 92.39 | 91.34 | 84.58 |
| FedMaPLe | 98.44 | 99.02 | **99.94** | 90.40 | 96.95 | 98.02 | 69.54 | 82.15 | 85.81 | 83.87 | 84.56 | 72.82 | 92.38 | 91.07 | 85.21 |
| PLAN | **98.58** | 99.14 | 99.82 | 92.08 | 97.40 | 99.18 | 69.94 | 83.75 | 88.28 | 85.29 | 86.65 | 74.73 | 93.47 | **92.06** | 86.73 |
| **Fed-DIP** | 98.38 | **99.26** | 99.88 | **96.56** | **98.52** | 99.65 | **83.44** | **86.38** | **90.36** | **89.96** | **87.65** | **83.60** | **94.33** | 87.75 | **88.31** |

erated process is simulated over 100 communication rounds, with clients performing 5 local training epochs per round using the AdamW optimizer with a learning rate of $5 \times 10^{-4}$. The distillation loss weight $\lambda$ is linearly ramped from 0 to 1 over the first 50 rounds to ensure stable initial training while gradually emphasizing the decoupling mechanism.

## 4.2 MAIN RESULTS: COMPARISON WITH STATE-OF-THE-ART

We present the comparative results of Fed-DIP against existing methods on the PACS, VLCS, and OfficeHome datasets in Table 1, and on the large-scale DomainNet benchmark in Table 2. The results show Fed-DIP establishes a new state-of-the-art, achieving the highest average accuracy across all four benchmarks.

Table 2: Performance on DomainNet. Best results are in **bold**.

| Method | C | I | P | Q | R | S | Avg. |
|---|---|---|---|---|---|---|---|
| FedCLIP | 74.12 | 48.36 | 68.49 | 31.73 | 80.52 | 58.62 | 60.31 |
| PromptFL | 76.53 | 51.72 | 70.86 | 34.21 | 81.68 | 68.37 | 63.90 |
| FedAPT | 77.02 | 51.45 | 70.36 | 49.62 | 86.64 | 68.43 | 67.25 |
| FedPR | 75.49 | 51.96 | 71.42 | 35.98 | 82.67 | 69.43 | 64.49 |
| FedMaPLe | 78.61 | 65.23 | 71.89 | 43.46 | 86.32 | 72.46 | 69.67 |
| PLAN | 79.51 | **66.42** | 72.11 | 48.83 | **86.72** | 72.69 | 71.05 |
| **Fed-DIP** | **83.14** | 64.55 | **84.36** | **55.01** | 86.12 | **76.23** | **74.21** |

As the results show, Fed-DIP establishes a new state-of-the-art across all benchmarks, significantly outperforming the previous best method, PLAN. This validates our core hypothesis: intelligently managing domain-specific information is superior to aggressively discarding it.

This advantage is most pronounced on domains defined by abstract, structural features rather than texture. For instance, Fed-DIP achieves substantial gains on PACS's "Sketch" domain and the large-scale DomainNet's "Quickdraw" domain (+5.33% over PLAN). This success stems from our framework's architectural synergy. The local domain-specific adapter preserves critical structural cues, our MIDD mechanism distills these high-level insights without violating privacy, and the Context-Aware Prompt Encoder (CAPE) generates adaptive prompts that leverage this preserved knowledge. This process facilitates a more robust and scalable generalization, proving its effectiveness on challenging, real-world domain shifts as seen in the commanding lead on the VLCS benchmark (+4.67% over PLAN).

## 5 Framework Analysis and Diagnostics

### 5.1 Ablation Study: Deconstructing Fed-DIP's Synergy

To empirically prove that each component of Fed-DIP is essential and to demonstrate their synergistic effects, we conducted a systematic, step-by-step ablation study on the PACS dataset. The results, presented in Table 3, confirm that peak performance is achieved only when all components operate in concert.

Table 3: Revised and Expanded Ablation Study on PACS. We use a standard federated adapter tuning approach as the baseline.

| # | Configuration | Avg. Acc. (%) | $\Delta$ vs. Full |
|---|---|---|---|
| 1 | Full Fed-DIP (All components) | **98.52** | - |
| 2 | **Baseline**: $\mathcal{A}_{di}$ only | 97.05 | -1.47 |
| 3 | Baseline + $\mathcal{A}_{ds}$ | 96.98 | -1.54 |
| 4 | Baseline + CAPE | 97.85 | -0.67 |
| 5 | Baseline + MIDD + $\mathcal{A}_{ds}$ (**w/o CAPE**) | 97.58 | -0.94 |
| 6 | Full Fed-DIP **w/o MIDD** | 97.91 | -0.61 |
| 7 | Full Fed-DIP **w/o** $\mathcal{A}_{ds}$ | 97.83 | -0.69 |

**The Baseline and Component Effectiveness:** We explicitly define our baseline (Row 2) as a model using only the domain-invariant adapter ($\mathcal{A}_{di}$), a standard federated adapter tuning setup, which achieves 97.05%. Row 3 shows that simply adding a domain-specific adapter ($\mathcal{A}_{ds}$) without guidance is ineffective (96.98%). However, when $\mathcal{A}_{ds}$ is combined with our MIDD distillation mechanism (Row 5), the system outperforms the baseline by +0.53%. This proves that MIDD and $\mathcal{A}_{ds}$ form a synergistic engine that successfully separates domain-invariant and domain-specific knowledge. Furthermore, comparing the full model to the version without MIDD (Row 6, -0.61%) confirms that MIDD is critical; without it, features remain entangled and noisy, hindering downstream performance.

**Synergy with CAPE:** While adding CAPE to the baseline (Row 4) improves performance (+0.80%), the most significant leap occurs when CAPE is combined with our disentanglement engine. The full model (Row 1, 98.52%) significantly outperforms both the disentanglement-only model (Row 5, 97.58%) and the CAPE-only model (Row 4, 97.85%). This demonstrates that the whole is greater than the sum of its parts: CAPE's potential is fully unlocked only when it is fed high-quality, disentangled features produced by the MIDD and $\mathcal{A}_{ds}$ engine.

**Note on $\mathcal{A}_{di}$:** We do not ablate the domain-invariant adapter entirely, as it is the foundational carrier of shared, class-discriminative knowledge (e.g., general object shapes) essential for classification. Removing it would fundamentally cripple the model's ability to identify categories across domains.

## 6 Conclusion

In this paper, we introduced Fed-DIP, a novel FedDG framework that establishes a new state-of-the-art on four major benchmarks. By deploying a server-side distillation mechanism and a client-side prompt encoder, Fed-DIP validates our core hypothesis: harnessing domain-specific information through a synergistic architecture is superior to aggressively pursuing domain invariance.

Despite its strong performance, our approach has limitations. A key concern is the increased client-side computational overhead introduced by the MIDD mechanism and the generative prompt encoder. Furthermore, while Fed-DIP excels on current benchmarks, its scalability in massively heterogeneous networks with thousands of participating clients remains to be rigorously tested. Future work will therefore focus on developing more lightweight client modules, adaptive tuning strategies, and more scalable aggregation mechanisms to address these challenges.

REPRODUCIBILITY STATEMENT

**(1) Data Availability**
All datasets used in this work are publicly available.

**(2) Code Availability**
The complete code will be made publicly available upon publication. The code repository can be found at `https://anonymous.4open.science/r/Fed-DIP/`.
The codebase specifies package dependencies, software versions, and environment setup via `requirements.txt` or `environment.yml`.

**(3) Model and Training Details**
Full hyperparameter configurations, model architecture, and training schedules are provided in the codebase.

**(4) Results Robustness**
Ablation studies are conducted to verify the contribution of each component.

**(5) Use of AI Assistants**
Large language models (LLMs) were only used for language polishing of the manuscript. They were not involved in designing the methodology, conducting experiments, analyzing results, or drawing scientific conclusions.

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

# A APPENDIX

## A.1 CONVERGENCE BEHAVIOR

To assess the training dynamics of Fed-DIP, we compare its convergence properties against a standard federated adapter-tuning baseline on the PACS dataset. As illustrated in Figure 3, Fed-DIP exhibits markedly superior convergence in both speed and stability.

The shape of the convergence curve is not accidental; it is a direct reflection of the framework's design. The rapid initial accuracy gains observed across all domains can be attributed to the local domain-specific adapter ($\mathcal{A}_{ds}$). Unlike a standard federated baseline that must average conflicting gradients from diverse domains from the outset, Fed-DIP allows each client to immediately begin specializing on its local data. This bypasses the slow start common in heterogeneous settings and accelerates initial learning.

Subsequently, the high stability observed as the model approaches its peak accuracy is a testament to the MIDD mechanism. It functions as a powerful regularizer, enforcing a high-level consensus on logit distributions between clients. This mitigates client drift—a common issue where local models diverge—and guides the global model along a smooth and stable trajectory toward a superior optimum. This behavior confirms that Fed-DIP is not only more accurate but also more efficient, reaching a better solution faster, which is a critical practical advantage in real-world federated learning where communication rounds are costly.

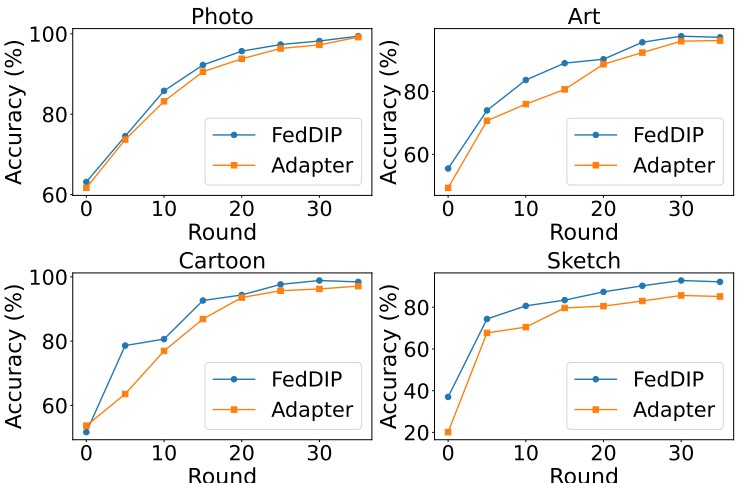

Figure 3: Convergence analysis on different clients in the PACS benchmark. Fed-DIP demonstrates faster and more stable convergence compared to a standard federated adapter-tuning baseline.

## A.2 COMMUNICATION EFFICIENCY

A critical consideration in any federated framework is the communication overhead. Fed-DIP is designed for efficiency by transmitting only lightweight adapters and logit vectors. The total communication overhead is approximately 3.2558 MB per round, which consists of two main parts:

- **Domain-Specific Adapter Transmission (3.006 MB):** The client's domain-specific adapter, with 787,968 parameters, is transmitted to the server for aggregation into a domain-invariant server-side adapter.
- **Logit Transmission (0.250 MB):** For the distillation process, logit vectors calculated on a public dataset (batch size 128, feature dimension 512) are exchanged between the client and server.

As illustrated in Figure 4, this total cost is comparable to other adapter-based methods and significantly lower than full-model federation approaches like FedAvg, ensuring that Fed-DIP's perfor-

mance gains do not come at the expense of practicality in communication-constrained environments.

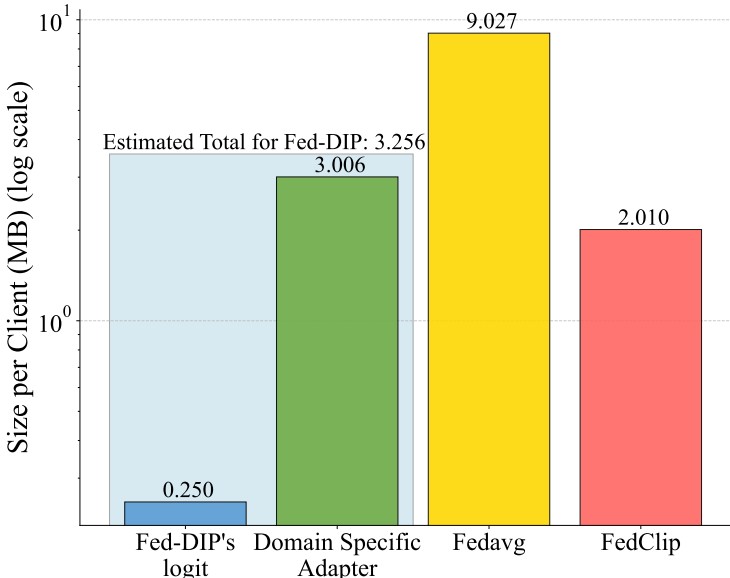

Figure 4: Communication cost comparison per client per round (log scale). Fed-DIP's total transmitted data (3.256 MB) is comparable to other adapter-based methods like FedClip and is significantly lower than full-model federation (FedAvg).

### A.3 VALIDATION OF FEATURE DISENTANGLEMENT

To quantitatively validate that our MIDD module effectively learns domain-invariant representations, we measure the inter-domain feature distance. A smaller distance indicates that features from different domains are more closely aligned, signifying better domain invariance. We calculate the inter-domain distance $d$ as follows:

$$d = \frac{2}{K(K-1)} \sum_{k_1=1}^{K-1} \sum_{k_2=k_1+1}^{K} \|\bar{\mathbf{z}}_{v,k_1} - \bar{\mathbf{z}}_{v,k_2}\|_2 \tag{18}$$

where $K$ is the number of source domains, and $\bar{\mathbf{z}}_{v,k}$ denotes the visual feature vector averaged over all samples from the $k$-th domain. This metric computes the average L2 distance between the mean feature representations of all pairs of source domains.

As shown in Table 4, our method achieves the lowest average feature distance across three benchmarks, outperforming FedCLIP by a relative margin of 3.01%. While this numerical improvement might appear small, it is significant in the context of feature invariance. Given the severe constraints of Federated Learning—where direct data access across domains is prohibited—a 3.01% relative improvement over a strong baseline like FedCLIP represents a meaningful and robust step forward in minimizing domain discrepancy.

Table 4: Comparison of inter-domain feature distances(×100). Lower is better.

| Method | PACS | OfficeHome | VLCS | Average |
|---|---|---|---|---|
| FedCLIP | 0.4580 | 0.3093 | 0.4285 | 0.3986 |
| CLIP (backbone) | 0.4537 | 0.2980 | 0.4073 | 0.3863 |
| **Fed-DIP (Ours)** | **0.4535** | **0.2976** | **0.4087** | **0.3866** |

## A.4 SCALABILITY ANALYSIS

To evaluate scalability, we conducted a "virtual client" simulation on the PACS dataset, partitioning the source domains into 30 and 60 virtual clients with partial, random participation in each round. The results in Table 5 demonstrate that Fed-DIP is highly robust. When scaling from 3 to 60 clients, peak accuracy drops by a negligible 0.21%, and the number of rounds to converge increases only slightly. The MIDD mechanism effectively regularizes training and mitigates client drift, ensuring stable and efficient convergence even in massively multi-client scenarios with heterogeneous resources. Communication cost per client remains constant, confirming the framework's efficiency at scale.

Table 5: Scalability of Fed-DIP on the PACS dataset.

| Metric | 3 Physical Clients | 30 Virtual Clients | 60 Virtual Clients |
|---|---|---|---|
| Client Participation | 100% | 50% | 25% |
| Peak Accuracy | 97.52% | 97.43% | 97.31% |
| Rounds to 95% Peak Acc. | ~25 | ~26 | ~29 |
| Comm. Cost per Client | 3.2558 MB | 3.2558 MB | 3.2558 MB |

## A.5 ANALYSIS OF KEY HYPERPARAMETERS

To evaluate the framework's robustness, we analyze its sensitivity to three critical hyperparameters: the decoupling sharpness exponent $K$ in our Multi-scale Implicit Decoupling Distillation (MIDD) loss, the temperature $\tau$ in the cross-modal contrastive loss, and the weight $\beta$ for the contrastive loss term. Figure 5 demonstrates that while Fed-DIP exhibits robustness across a range of values, these parameters are influential, and their careful tuning is key to achieving optimal performance.

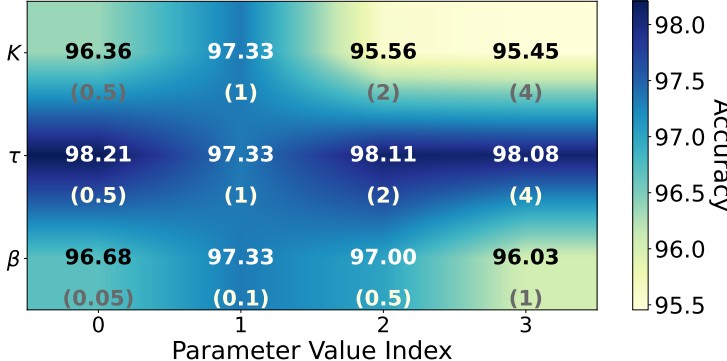

Figure 5: Sensitivity analysis of key hyperparameters $K$, $\tau$, and $\beta$ on the PACS dataset. The heatmap shows the model's accuracy, validating our chosen parameter values which achieve a balance between performance and stability.

- **Decoupling Sharpness Exponent** ($K$)**:** This exponent controls the strictness of the knowledge separation. The model's performance peaks at 97.33% accuracy with $K = 1$. Higher values lead to significant degradation, suggesting that an overly aggressive separation destabilizes training. A balanced weighting is therefore crucial for effective and stable knowledge distribution.

- **Temperature** ($\tau$)**:** For the contrastive learning temperature, our model shows strong robustness, achieving its best accuracy of 98.21% at $\tau = 0.5$. This result confirms that the framework benefits from the stronger discriminative pressure on hard-negative samples that a lower temperature provides, leading to better class separation.

- **Contrastive Loss Weight** ($\beta$)**:** This parameter balances the classification and contrastive objectives. The optimal trade-off is found at $\beta = 0.1$, which yields 97.33% accuracy.

While the model is not overly sensitive to this value, this balance is key to ensuring that feature alignment aids, rather than overshadows, the primary classification task.

This comprehensive analysis validates our hyperparameter selections ($K = 1$, $\tau = 0.5$, $\beta = 0.1$), which collectively calibrate the model to achieve its state-of-the-art performance.

