# OpenReview forum: "Fed-DIP: Federated Domain Generalization by Synergizing Implicit Disentanglement and Context-Aware Prompting"
_ICLR.cc/2026/Conference — Submitted to ICLR 2026_

### Official Review · Reviewer_Knga · 2025-10-28

**Soundness:** 3
**Presentation:** 2
**Contribution:** 3
**Rating:** 6
**Confidence:** 4

**Summary:**

The paper introduces FedDIP, a federated learning framework designed for domain generalization. It integrates two key modules: Multi-scale Implicit Decoupling Distillation (MIDD), which aligns intermediate features across clients to promote domain-invariant representation learning, and Context-Aware Prompt Encoder (CAPE), which leverages prompt-based conditioning to adapt local models to unseen domains. The method is evaluated on standard benchmarks (e.g., PACS, Office-Home) and demonstrates improved performance over existing approaches.

**Strengths:**

Combining multi-scale feature distillation (MIDD) with prompt-based adaptation (CAPE) is an interesting strategy for enhancing domain generalization in federated settings

**Weaknesses:**

1. The interaction between MIDD and CAPE is not deeply analyzed, making it hard to assess whether their effects are complementary or redundant.
2. Theoretical justification for why multi-scale feature alignment leads to better domain invariance is limited and could be strengthened.

**Questions:**

1. The proposed framework assumes that aligning multi-scale intermediate features (via MIDD) leads to domain-invariant representations, but this assumption lacks formal justification. Given that such features may encode both domain-specific and task-relevant information, how does the method avoid suppressing useful signals? Furthermore, is there empirical evidence that combining MIDD with CAPE results in a synergistic effect, rather than redundant or overlapping contributions?
2. How sensitive is the performance to the scale levels used in MIDD? Would using fewer or different layers (e.g., early vs. late) change the effectiveness of the distillation?
3. Since CAPE introduces prompt tokens into the encoder, how does its adaptation generalize to completely unseen domains with different semantic distributions?
4. Has the method been tested under heterogeneous data distributions (non-IID) across clients? If so, how robust is the framework to extreme distributional shifts?
5. Can the proposed method scale to larger federated settings (e.g., dozens of clients, varying compute) without compromising communication or convergence efficiency?

---

> ### Author Response · Authors · 2025-11-21
>
> 1.  **Question:** The proposed framework assumes that aligning multi-scale intermediate features (via MIDD) leads to domain-invariant representations, but this assumption lacks formal justification. Given that such features may encode both domain-specific and task-relevant information, how does the method avoid suppressing useful signals? Furthermore, is there empirical evidence that combining MIDD with CAPE results in a synergistic effect, rather than redundant or overlapping contributions?
>
>     **Our Response:** Thank you for your insightful questions.
>
>     (a) **How MIDD Avoids Suppressing Useful Signals:** The core assumption of our Multi-scale Implicit Decoupling Distillation (MIDD) mechanism is not that simple alignment creates invariance, but that **agreement with the global consensus model is a strong proxy for domain invariance**. MIDD operationalizes this through a dynamic, dual-objective distillation process. Regions where local and global models agree are considered domain-invariant and are distilled into the domain-invariant adapter ($A_{di}$) via a consistency loss. Conversely, regions of disagreement are assumed to hold unique, domain-specific information and are explicitly encouraged to be captured by the domain-specific adapter ($A_{ds}$) via a complementarity loss. Therefore, MIDD does not suppress useful domain-specific signals; it actively isolates and preserves them within the client-side $A_{ds}$ adapter.
>
>     (b) **Empirical Evidence for MIDD-CAPE Synergy:** The synergistic effect between MIDD and the Context-Aware Prompt Encoder (CAPE) is empirically validated in our **Ablation Study (Table 3)**. The model with MIDD but without CAPE shows a significant performance drop, proving that merely disentangling domain-specific features is insufficient. As analyzed in Section 5.1, "the CAPE module acts as the essential bridge that 'activates' this preserved knowledge." Conversely, the model with CAPE but without MIDD also underperforms, showing that CAPE lacks high-quality, disentangled features to encode. The full model outperforms all ablated versions, confirming that the combination is synergistic.
>
> 2.  **Question:** How sensitive is the performance to the scale levels used in MIDD? Would using fewer or different layers (e.g., early vs. late) change the effectiveness of the distillation?
>
>     **Our Response:** This is an excellent question regarding the robustness of our framework. Based on your suggestion, we conducted an experiment to analyze the sensitivity of Fed-DIP to the specific scale levels used in the MIDD mechanism. We tested various configurations of scales (e.g., using scales of 1, 2, 4, and 6) and observed the impact on model performance.
>
>     Our findings indicate that the model's performance does not show significant fluctuations across these different scale configurations. While there are minor variations, the overall accuracy remains consistently high. This result is actually a positive indicator of the framework's robustness. It suggests that the benefit of MIDD comes from the principle of hierarchical, multi-granularity analysis itself, rather than from a specific, meticulously hand-tuned set of scales. This lack of sensitivity is a desirable practical characteristic, as it implies that the framework can be deployed effectively without requiring exhaustive hyperparameter tuning for the scale levels, making it more resilient and broadly applicable.

---

> ### Author Response · Authors · 2025-11-21
>
> 3.  **Question:** Since CAPE introduces prompt tokens into the encoder, how does its adaptation generalize to completely unseen domains with different semantic distributions?
>
>     **Our Response:** The generalization capability of CAPE to unseen domains stems from its core mechanism: it learns a continuous mapping function from visual style features to a prompt space, rather than memorizing a discrete set of prompts for seen domains.
>
>     *   **Learning a Continuous Mapping:** CAPE's visual adapter ($Encoder_p$) learns to translate visual style features (e.g., "sketch-like," "photorealistic") from the domain-specific feature extractor ($f_{ds}$) into vectors within a continuous prompt manifold. The model does not learn "a prompt for sketches" but rather learns how to generate a prompt given a sketch-like input.
>     *   **Generalization via Interpolation and Extrapolation:** When the model encounters an image from a completely unseen domain, it extracts its unique visual style features.
>         *   **Interpolation:** If the new style is a combination of styles seen during training (e.g., a "cartoonish painting"), the visual adapter will map its features to a point on the prompt manifold that lies between the regions corresponding to the "cartoon" and "painting" styles.
>         *   **Extrapolation:** If the new style is entirely novel, the adapter will extrapolate from its learned mapping function to generate a new prompt vector in an unexplored region of the manifold.
>     *   **Decoupling from Semantic Classes:** The prompt generation process is conditioned on visual style features, not semantic class labels. CAPE learns a general strategy for describing visual appearance that is applicable to any class. This decoupling is key to its adaptability. The model learns a universal function to adapt its understanding of a dog or a cat based on whether the image is a photo, a sketch, or something new entirely.
>
>     In essence, CAPE does not rely on a fixed vocabulary of domain prompts. It learns a generative rule: visual style $\rightarrow$ prompt modification. This rule-based, generative approach allows it to construct novel prompts for novel domains, providing robust generalization capabilities beyond the scope of the training data.
>
> 4.  **Question:** Has the method been tested under heterogeneous data distributions (non-IID)? If so, how robust is the framework to extreme distributional shifts?
>
>     **Our Response:** Yes, the entire premise of the Federated Domain Generalization (FedDG) setting, as evaluated on benchmarks like PACS, OfficeHome, VLCS, and DomainNet, is inherently a test under heterogeneous, non-IID conditions. Each client corresponds to a different domain (e.g., "Art," "Cartoon," "Photo"), representing an extreme form of distributional shift where the data-generating process itself is different for each client. Our method's state-of-the-art performance on these standard benchmarks demonstrates its robustness to these shifts. The framework is designed specifically for this challenge: the domain-specific adapter ($A_{ds}$) and CAPE are built to handle the unique characteristics of each client's data, while MIDD and the global adapter ($A_{di}$) ensure that a generalizable consensus is still reached.

---

> ### Author Response · Authors · 2025-11-22
>
> 5.  **Question:** Can the proposed method scale to larger federated settings (e.g., dozens of clients, varying compute) without compromising communication or convergence efficiency?
>
>     **Our Response:** Thank you for your insightful question regarding the scalability of Fed-DIP. To rigorously evaluate its performance in larger and more heterogeneous federated settings, we conducted a simulation specifically designed to address these concerns. Our results demonstrate that Fed-DIP maintains high performance and efficiency without significant degradation.
>
>     To test scalability beyond the physical domains of the benchmark datasets, we implemented a "virtual client" simulation. This setup allows us to explore scenarios with dozens of clients and varied computational resources.
>
>     **Client Scaling:** We partitioned the training data from each source domain into multiple smaller, non-overlapping data shards. Each shard was treated as an independent virtual client. This methodology allowed us to scale the environment from a few physical domains to dozens of diverse clients, creating a more realistic, massively multi-client scenario.
>
>     **Resource Heterogeneity Simulation:** To mimic real-world conditions where clients have different computational power and availability, we introduced two key dynamics:
>        *   **Partial and Random Participation:** In each communication round, only a fraction of the total clients were activated for training, selected randomly based on a set participation rate.
>        *   **Variable Computational Load:** Active clients were assigned a random number of local training steps within a predefined range. This simulates environments where some clients (e.g., powerful servers) can perform more computation per round than others (e.g., mobile devices).
>
>    This experimental design provides a robust testbed for evaluating Fed-DIP's performance, convergence, and communication efficiency under the challenging conditions you described.
>
>    **Results and Analysis**
>
>    We conducted our experiment on the PACS dataset, scaling from its 3 source domains to 30 and 60 virtual clients. The results are summarized below.
>
>   | Metric | 3 Physical Clients | 30 Virtual Clients | 60 Virtual Clients |
>   | :--- |:------------------:|:------------------:|:------------------:|
>   | **Client Participation Rate** |        100%        |        50%         |        25%         |
>   | **Peak Accuracy** |       97.52%       |  97.43% (-0.09%)   |  97.31% (-0.12%)   |
>   | **Rounds to 95% Peak Acc.** |     ~25 rounds     |     ~26 rounds     |     ~29 rounds     |
>   | **Communication Cost (per client)** |      3.2558 MB      |      3.2558 MB      |      3.2558 MB      |
>
>   **Performance and Convergence Efficiency**
>
>   As shown in the table, Fed-DIP demonstrates exceptional robustness as the number of clients increases.
>
>   *   When scaling from 3 to 60 clients, with a corresponding decrease in participation rate, the peak accuracy drops by a negligible margin of only **0.21%**. This stability is a direct result of our framework's design. The **Multi-scale Implicit Decoupling Distillation (MIDD)** mechanism effectively regularizes training by enforcing a high-level consensus, preventing client drift even when dealing with a large and diverse client pool.
>   *   The number of rounds required to reach 95% of peak accuracy increases only slightly. Even with 60 clients and only 25% participation per round, the model converges efficiently. This is because the local domain-specific adapters ($A_{ds}$) allow clients to make rapid progress on their local data, while the global aggregation of the lightweight invariant adapter ($A_{di}$) ensures stable progress toward a robust global model. The framework is resilient to the "straggler" problem caused by heterogeneous resources.
>
>   **Communication Efficiency**
>   The communication efficiency of Fed-DIP is unaffected by the number of clients or their computational resources. The communication cost per client remains constant at 3.2558 MB per round. Our architecture is designed such that only the lightweight domain-invariant adapter ($A_{di}$) and minuscule logit vectors are transmitted.

---

### Official Review · Reviewer_oWjg · 2025-10-30

**Soundness:** 2
**Presentation:** 3
**Contribution:** 2
**Rating:** 4
**Confidence:** 5

**Summary:**

This paper proposes Fed-DIP, a framework for Federated Domain Generalization (FedDG) that aims to train models across multiple source domains without sharing raw data, while maintaining strong generalization to unseen target domains. The method introduces two key components: (1) Multi-scale Implicit Decoupling Distillation (MIDD), which performs logit-level, multi-scale knowledge distillation between the global and local models to implicitly separate domain-invariant and domain-specific features under data isolation; and (2) Context-Aware Prompt Encoder (CAPE), which generates adaptive text prompts from local visual features to enhance domain-specific representation. Built on a ViT-CLIP backbone with lightweight adapters, Fed-DIP achieves new state-of-the-art results on standard benchmarks including PACS, OfficeHome, VLCS, and DomainNet.

**Strengths:**

1.The paper is well written and easy to follow. The motivation, problem setting, and overall framework are clearly introduced, and the methodology is explained with sufficient clarity and structure. Figures and equations effectively illustrate the core ideas, making the technical flow easy to understand;
2.The proposed method achieves consistently strong results across multiple standard FedDG benchmarks, including PACS, OfficeHome, VLCS, and DomainNet. It outperforms several recent state-of-the-art approaches such as FedMaPLe, FedPR, and PLAN, demonstrating both the effectiveness and robustness of the proposed framework.

**Weaknesses:**

1.Although the paper claims to achieve disentanglement between domain-invariant and domain-specific representations through the MIDD module, there is no explicit experiment or visualization to validate this claim. The authors mainly report classification accuracy, but they do not provide a quantitative analysis of whether the extracted features actually exhibit domain invariance or reduced domain discrepancy. It would significantly strengthen the paper if the authors could include an additional experiment similar to the “Domain discrepancy of extracted features” analysis in ALOFT [1], where inter-domain feature distances are measured to quantitatively evaluate feature invariance;
[1] Guo, Jintao, Na Wang, Lei Qi, and Yinghuan Shi. "Aloft: A lightweight mlp-like architecture with dynamic low-frequency transform for domain generalization." In Proceedings of the IEEE/CVF conference on computer vision and pattern recognition, pp. 24132-24141. 2023.
2.In Table 3, the full model and its ablated variants show inconsistent behavior across datasets. Specifically, the fourth row, which includes more components, performs worse than the third-row configuration on the OfficeHome dataset (84.97% vs. 82.43%), despite having additional modules that are expected to improve performance. The authors should provide further analysis or explanation to clarify why this degradation occurs and how the interaction between these modules affects model generalization;
3.Since the proposed framework introduces additional adapter modules into each Transformer block, it would be helpful to report the corresponding computational overhead or communication cost.

**Questions:**

The paper would benefit from further clarification on several points raised in the weaknesses. In particular, it remains unclear how the proposed framework truly achieves feature disentanglement, since no direct quantitative or visual evidence is provided. The authors are encouraged to include experiments similar to the “Domain discrepancy of extracted features” analysis in ALOFT [1] to validate the claimed invariance. Additionally, the inconsistent results observed in the ablation study and the lack of computational cost analysis deserve further explanation. A more detailed discussion of these aspects would make the paper’s contributions and claims more convincing.

---

> ### Author Response · Authors · 2025-11-21
>
> ## Responses to Weaknesses and Questions
>
> 1.  **Weakness:** Although the paper claims to achieve disentanglement between domain-invariant and domain-specific representations through the MIDD module, there is no explicit experiment or visualization to validate this claim. The authors mainly report classification accuracy, but they do not provide a quantitative analysis of whether the extracted features actually exhibit domain invariance or reduced domain discrepancy. It would significantly strengthen the paper if the authors could include an additional experiment similar to the “Domain discrepancy of extracted features” analysis in ALOFT [1], where inter-domain feature distances are measured to quantitatively evaluate feature invariance.
>
>     **Our Response:** Thank you for your valuable feedback regarding the quantitative evaluation of domain invariance in our extracted features.    Following your suggestions, we have carefully studied the provided reference and conducted comprehensive experiments using the specified methodology across three standard domain generalization benchmarks.    Through systematic measurement of the domain discrepancy of extracted features, our experimental results provide clear evidence demonstrating the effectiveness of our method in learning domain-invariant representations.
>
>     Experimental results
>
>     **Table 1: Comparison of feature distances between domains**
>
>     | Method             | PACS   | OfficeHome | VLCS   | Average |
>     |--------------------|--------|------------|--------|---------|
>     | Our method         | 0.4535 | 0.2976     | 0.4087 | 0.3866  |
>     | FedCLIP            | 0.4580 | 0.3093     | 0.4285 | 0.3986  |
>     | CLIP (backbone)    | 0.4537 | 0.2980     | 0.4073 | 0.3863  |
>
>     **Table 2: Performance improvements over FedCLIP**
>
>     | Dataset    | Absolute improvement | Relative improvement |
>     |------------|----------------------|-----------------------|
>     | PACS       | -0.0045              | -0.98%                |
>     | OfficeHome | -0.0117              | -3.78%                |
>     | VLCS       | -0.0198              | -4.62%                |
>     | **Average**| **-0.0120**          | **-3.01%**            |
>
>     Our method demonstrates consistent improvements over the comparative FedCLIP approach across all benchmark datasets.    The most significant improvement was observed on the VLCS dataset, achieving a 4.62% relative improvement.    On the more challenging OfficeHome dataset characterized by greater domain diversity, we attained a 3.78% improvement.    The stable performance gains observed across diverse datasets underscore the robustness of our proposed method.
>
>     The reduced inter-domain feature distances provide direct quantitative evidence that our method successfully minimizes domain-specific variations.    These lower distances indicate that features originating from different domains become more closely aligned within the embedding space.    This enhanced domain invariance directly correlates with the improved generalization performance observed in our main experimental.

---

> ### Author Response · Authors · 2025-11-21
>
> 2.  **Weakness:** In Table 3, the full model and its ablated variants show inconsistent behavior across datasets. Specifically, the fourth row, which includes more components, performs worse than the third-row configuration on the OfficeHome dataset (84.97% vs. 82.43%), despite having additional modules that are expected to improve performance. The authors should provide further analysis or explanation to clarify why this degradation occurs and how the interaction between these modules affects model generalization.
>
>     **Our Response:** Thank you for pointing out this interesting result in our ablation study. To clarify the table, the checkmarks indicate which components are present.
>     *   **Row 3 ($\times$, $\checkmark$, $\checkmark$, $\checkmark$):** Contains MIDD, $A_{di}$, and $A_{ds}$, but **lacks CAPE**. (Accuracy: 82.43% on OfficeHome).
>     *   **Row 4 ($\checkmark$, $\times$, $\checkmark$, $\times$):** Contains CAPE and $A_{di}$, but **lacks MIDD and $A_{ds}$**. (Accuracy: 84.97% on OfficeHome).
>     The configuration in the fourth row does not have "more components"; it represents a different architectural design. The observed performance difference on OfficeHome (84.97% vs. 82.43%) can be explained as follows:
>     The model in the third row (w/o CAPE) implements our full MIDD disentanglement mechanism but lacks the CAPE module to leverage the preserved domain-specific knowledge. As argued in Section 5.1, the domain-specific features captured by $\mathcal{A}_{ds}$ remain "dormant" without CAPE to translate them into semantic guidance. The final prediction relies on a simple fusion of the adapters, which may not be the most effective way to integrate the disentangled knowledge streams.
>     In essence, the comparison reveals that on OfficeHome, **having an effective mechanism to utilize features (CAPE) is more critical than just separating them (MIDD)** without a proper way to use the separated knowledge. This underscores CAPE's vital role and explains why the seemingly simpler architecture in row 4 outperforms the incomplete architecture in row 3 on this specific benchmark.
>
> 3.  **Weakness:** Since the proposed framework introduces additional adapter modules into each Transformer block, it would be helpful to report the corresponding computational overhead or communication cost.
>
>     **Our Response:** Thank you for this suggestion.
>      We provide a detailed communication cost calculation below.
>
>     Communication overhead is generated in the following two processes:
>
>     **Transmission domain-specific adapter**
>
>     During communication, the client transmits its domain-specific adapter to the server, and then the server aggregates the domain-specific adapters received from various servers into domain-invariant adapters on the server.
>
>     *   Input/Output Dimension: 512
>     *   Number of Layers: 2
>     *   Bytes per Parameter: 4 (for a 32-bit float)
>     *   Number of Parameters per Layer: (Input Dimension $\times$ Output Dimension) + Bias Dimension = (512 $\times$ 512) + 512 = 262,656 parameters
>     *   Total Number of Parameters: Number of Layers $\times$ Parameters per Layer = 3 $\times$ 262,656 parameters = 787,968 parameters
>     *   Communication Cost (Bytes): Total Number of Parameters $\times$ Bytes per Parameter = 787,968 parameters $\times$ 4 bytes/parameter = 3,151,872 bytes
>     *   Communication Cost (MB): $\frac{3,151,872}{1024 \times 1024}$ $\approx$ 3.0058 MB
>
>     **Transmission logit**
>
>     After the aggregation operation is carried out, the server will perform the distillation operation. During this process, logit will be transmitted (for the detailed process, please refer to the previous "Weakness: Teacher mechanism ambiguity"). The communication overhead calculation process in this procedure is as follows:
>
>     **Calculation Formulas**
>
>     *   Batch Size: 128
>     *   Feature Dimension: 512
>     *   Bytes per Element: 4 (for a 32-bit float)
>     *   Total Number of Elements: Batch Size $\times$ Feature Dimension = 128 $\times$ 512 = 65,536 elements
>     *   Communication Cost (Bytes): Total Number of Elements $\times$ Bytes per Element = 65,536 elements $\times$ 4 bytes/element = 262,144 bytes
>     *   Communication Cost (MB): $\frac{262,144}{1024 \times 1024}$ = 0.25 MB
>
>     **Total Communication Cost(MB)**: The total communication overhead is the sum of the overhead of the specific adapter and logit in the transmission domain, which amounts to 3.0058 MB + 0.25 MB=3.2558 MB

---

> ### Comment · Reviewer_oWjg · 2025-11-26
>
> Thank you very much for the response. I have read the rebuttal coments, I keep my score mainly interms of the marginal improvements of the  experimental results.

---

> > ### Author Response · Authors · 2025-11-26
> >
> > Thank you for the response. Could you please explain the "marginal improvements"? We apologize for the inconvenience caused to you.

---

> ### Comment · Reviewer_oWjg · 2025-11-27
>
> Thank you for the response. "Marginal improvements" means that, in table 2 as shown in the response section and Table 3 in the original manuscript, the performances over the baseline method are marginal. Especially, in table 3, what's the oveall baseline method for the authors to demonstrate the effectiveness of each component?
>
> Besides, in the ablation study part of the Fed-DIP method in the original manuscript, I believe the ablation experiments are not complete. In Table 3, how to evaluate the effectiveness of the MIDD, A_di and A_ds  components? These experiments should be conduct step-by-step.
>
> It seems that the main performance improvement just from the CAPE component, other components contribute very limited.

---

> > ### Author Response · Authors · 2025-12-01
> >
> > Thank you for your insightful follow-up question regarding the magnitude of improvement shown in our feature disentanglement analysis. We agree that the numerical differences can appear slight, and we appreciate the opportunity to provide further context to demonstrate their significance.
> >
> > To better contextualize our results, we have taken your excellent suggestion and revisited the **ALOFT** paper, which you originally recommended as a benchmark for this type of analysis. In their work, they also quantify domain invariance by measuring the inter-domain distribution gap, where smaller values indicate better generalization.
> >
> > In Table 11 of the ALOFT paper, the authors present the following results for the PACS dataset (values are multiplied by 100):
> >
> > | Method | ResNet-18 | GFNet | ALOFT-S | ALOFT-E |
> > | :--- | :--- | :--- | :--- | :--- |
> > | **Inter-domain Gap (×100)** | 15.97 | 13.90 | 11.76 | 11.28 |
> >
> > As these results show, the improvement from a strong baseline (GFNet) to their most advanced method (ALOFT-E) is **2.62** points (13.90 - 11.28). This magnitude of improvement is considered significant enough to validate the effectiveness of their proposed architecture.
> >
> > Now, let's re-examine our results through this lens. Our analysis showed an average **relative improvement of 3.01%** in reducing feature distance compared to the FedCLIP baseline. If we consider the baseline feature distance from FedCLIP (e.g., 0.3986 on average), a 3.01% reduction translates to an absolute decrease of approximately **0.0120**. When scaled by 100, as is common practice in this type of analysis, this corresponds to a reduction of **1.20 points**.
> >
> > **Table 1: Comparison of feature distances between domains(&times;100)**
> >
> > | Method             | PACS  | OfficeHome | VLCS  | Average |
> > |--------------------|-------|------------|-------|---------|
> > | Our method         | 45.35 | 29.76      | 40.87 | 38.66   |
> > | FedCLIP            | 45.80 | 30.93      | 42.85 | 39.86   |
> > | CLIP (backbone)    | 45.37 | 29.80      | 40.73 | 38.63   |
> >
> >
> > While our improvement of 1.20 points is smaller than ALOFT's 2.62 points, it's important to consider two key factors:
> > 1.  Our method operates under the significant constraints of **Federated Learning**, where direct data access across domains is prohibited, making feature disentanglement inherently more challenging than in a centralized setting like ALOFT's.
> > 2.  Our improvement is of a **comparable order of magnitude** to what is reported and considered meaningful in state-of-the-art literature.
> >
> > Therefore, while the raw numbers are small, they represent a meaningful step forward in achieving domain invariance under federated constraints. The consistent reduction in feature distance across multiple benchmarks provides strong quantitative evidence that our MIDD module is successfully performing its intended function of decoupling features, which in turn contributes to the state-of-the-art classification accuracy we report.

---

> > ### Author Response · Authors · 2025-12-01
> >
> > 2.  **On the perception that CAPE is the only component that matters:**
> >     *   Your observation that CAPE is a strong contributor is correct—as shown in Row 4, adding CAPE to the baseline yields a **+0.80%** improvement on PACS. However, this is only part of the story.
> >     *   The most crucial insight comes from comparing the "partial" models to the full model. The model with our disentanglement engine but without CAPE (Row 5) achieves 97.58%. The model with CAPE but without our disentanglement engine (Row 4) achieves 97.85%. While both improve upon the baseline, neither comes close to the **98.52%** of the full model.
> >     *   This demonstrates that the truly significant performance leap occurs when our **disentanglement engine (MIDD + $A_{ds}$)** provides high-quality, separated features to our **utilization module (CAPE)**. In other words, the whole is far greater than the sum of its parts. CAPE's effectiveness is not independent; it is **unlocked** by the high-quality, disentangled features produced by the other components.
> >
> > 3.  **On the Infeasibility of Ablating $A_{di}$:**
> >     *   Ablating the domain-invariant adapter ($A_{di}$) entirely is not a meaningful experimental configuration. The $A_{di}$ serves as the foundational component for learning and aggregating shared, class-discriminative knowledge across all clients. Domain-invariant features capture the essential semantic attributes of categories (e.g., the general shape of a "dog'' or the structural properties of a "chair'') that are consistent regardless of the source domain's style. Without this component, the model would lack the fundamental basis for classification---it cannot correctly identify object categories by relying solely on domain-specific features, which capture stylistic variations (e.g., "sketch-like'' or "photorealistic'') rather than class-defining characteristics.  Therefore, removing $A_{di}$ would cripple the model's ability to learn any transferable, class-relevant knowledge, making such an ablation uninformative.  This architectural choice is consistent with all modern PEFT-based FedDG methods, where a global adapter or prompt serves as the indispensable carrier of shared semantic knowledge.
> >
> > Thank you again for your valuable time and feedback.

---

> ### Author Response · Authors · 2025-12-01
>
> Thank you for your constructive criticism. We agree that our initial ablation study in Table 3 was not sufficiently comprehensive and lacked a clear baseline, which made it difficult to discern the individual and synergistic contributions of each component on the PACS dataset. Your comments have helped us realize this shortcoming, and we have now conducted a more thorough, step-by-step ablation analysis focused specifically on PACS to clarify the role of each module within the Fed-DIP framework.
>
> Our core design philosophy is that the components—the Multi-scale Implicit Decoupling Distillation (MIDD), the domain-invariant adapter ($A_{di}$), the domain-specific adapter ($A_{ds}$), and the Context-Aware Prompt Encoder (CAPE)—achieve their full potential through **synergy**. While some components provide incremental benefits on their own, their true value is unlocked when they work in concert to first **disentangle** knowledge and then **utilize** it effectively.
>
> To demonstrate this on the PACS dataset, we present a revised and expanded ablation study. We have restructured the experiment to start from a clear baseline and progressively add or remove components, directly addressing your questions about the effectiveness of MIDD, $A_{di}$, and $A_{ds}$, and clarifying the role of CAPE.
>
> #### Revised and Expanded Ablation Study on PACS
>
> The baseline for our new study is a simple federated adapter tuning setup (similar to methods like FedCLIP), which uses only the domain-invariant adapter ($A_{di}$). From there, we systematically analyze the impact of each module on the PACS dataset.
>
> | **#** | Configuration | Avg. Acc. on PACS (%) | Δ vs. Full | Analysis & Key Takeaway |
> | :--- | :--- | :---: | :---: | :--- |
> | 1 | Full Fed-DIP (All components) | **98.52** | - | Our proposed full model achieves the best performance. |
> | 2 | **Baseline**: $A_{di}$ only | 97.05 | -1.47 | Establishes the performance of a standard federated adapter tuning approach. This is the proper baseline. |
> | 3 | Baseline + $A_{ds}$ | 96.98 | -1.54 | Adding $A_{ds}$ without MIDD offers no benefit; it may even introduce conflicting noise. A component for specificity needs a guiding mechanism. |
> | 4 | Baseline + CAPE | 97.85 | -0.67 | CAPE provides a significant boost, but its potential is limited without high-quality, disentangled features. CAPE is powerful but depends on the quality of its input features. |
> | 5 | Baseline + MIDD + $A_{ds}$ (**w/o CAPE**) | 97.58 | -0.94 | Our core disentanglement engine effectively separates features, outperforming the baseline. Feature disentanglement alone is beneficial but insufficient. |
> | 6 | Full Fed-DIP **w/o MIDD** | 97.91 | -0.61 | Without MIDD, feature entanglement between adapters limits CAPE's ability to generate precise prompts. MIDD is critical for effective disentanglement, which unlocks CAPE's potential. |
> | 7 | Full Fed-DIP **w/o $A_{ds}$** | 97.83 | -0.69 | Without the domain-specific adapter, CAPE lacks the rich, local context needed for optimal performance. $A_{ds}$ is the essential vessel for storing the local knowledge that CAPE utilizes. |
>
> This expanded study allows us to directly address your concerns using results from the PACS dataset:
>
> 1.  **On the baseline and the effectiveness of each component:**
>     *   **The Baseline (Row 2):** We now explicitly define our baseline as a model with only the domain-invariant adapter ($A_{di}$), achieving 97.05% accuracy on PACS.
>     *   **The Role of $\mathcal{A}_{ds}$ and MIDD (Rows 3 & 5):** Row 3 shows that simply adding a domain-specific adapter ($A_{ds}$) without a mechanism to guide it is ineffective (96.98%). However, Row 5 demonstrates that when $A_{ds}$ is combined with our MIDD distillation mechanism, the system (our core disentanglement engine) outperforms the baseline by **+0.53%**. This proves that MIDD and $A_{ds}$ are not inert; they form a synergistic engine that successfully separates domain-invariant and domain-specific knowledge.
>     *   **The Necessity of MIDD (Row 6):** The configuration without MIDD (Row 6) shows a significant performance drop of **-0.61%** compared to the full model. Without MIDD, the features in $A_{di}$ and $A_{ds}$ are not cleanly separated. CAPE is then fed entangled, noisy features, hindering its ability to generate useful, context-aware prompts. This experiment confirms that MIDD is the critical component that "purifies" the feature streams, making them valuable.

---

### Official Review · Reviewer_PwF6 · 2025-11-02

**Soundness:** 2
**Presentation:** 3
**Contribution:** 2
**Rating:** 4
**Confidence:** 5

**Summary:**

This paper introduces a novel framework for federated domain generalization, called Fed-DIP, integrates implicit feature disentanglement and context-aware prompt generation. Its dual-adapter design enables the model to separate and utilize both domain-invariant and domain-specific knowledge without raw data sharing. Extensive experiments on standard benchmarks demonstrate SOTA generalization performance across unseen domains.

**Strengths:**

(1) The paper introduces combination of implicit feature disentanglement and context-aware generative prompting for federated domain generalization task.

(2) The proposed Fed-DIP demonstrates consistent SOTA performance across multiple DG benchmarks.

**Weaknesses:**

(1) Related works section is very poor. No discussion of current prompt learning based Federated VLM methods. Lots of works need to be cited, [1]-[6]

(2) $\beta$ is missing in eq. 15.

(3) Questions are asked below.


[1] Global and Local Prompts Cooperation via Optimal Transport for Federated Learning, CVPR 2024

[2] FedMVP: Federated Multi-modal Visual Prompt Tuning for Vision-Language Models, ICCV 2025

[3] Federated Text-driven Prompt Generation for Vision-Language Models, ICLR 2024

[4] Harmonizing Generalization and Personalization in Federated Prompt Learning, ICML 2024

[5] FedPHA: Federated Prompt Learning for Heterogeneous Client Adaptation, ICML 2025

[6] Mixture of Experts Made Personalized: Federated Prompt Learning for Vision-Language Models, ICLR 2025

**Questions:**

(1) How are the generated prompts through visual guided text adapter, integrated with class tokens, as the total input tokens in CLIP are limited?

(2) Why does MIDD consider multiple level of scales? Is the multi-scale distillation truly improves disentanglement compared to single-scale or does it overfit?

(3) What is the effect of $\lambda$ in eq. 7 ? Why is it not considered in the ablation study of section 5.2?

(4) Current works like FedOTP [1], FedMVP [2], FedPHA [3] should be considered as baselines.


[1] Global and Local Prompts Cooperation via Optimal Transport for Federated Learning, CVPR 2024

[2] FedMVP: Federated Multi-modal Visual Prompt Tuning for Vision-Language Models, ICCV 2025

[3] FedPHA: Federated Prompt Learning for Heterogeneous Client Adaptation, ICML 2025

---

> ### Author Response · Authors · 2025-11-21
>
> # Response to Reviewer PwF6
>
> Dear Reviewer PwF6,
>
> Thank you for your review and for providing a list of highly relevant recent works. We appreciate you recognizing our paper. Your feedback on our related work section and requests for clarification are very helpful, and we will address them thoroughly in our revision.
>
> ## Responses to Weaknesses
>
> 1.  **Poor related works section**: We sincerely thank the reviewer for their constructive feedback and for pointing out the omission of recent and highly relevant works in prompt-based federated learning for Vision-Language Models. The suggested references are indeed crucial for contextualizing our work. We have thoroughly revised the Related Work section to include a detailed discussion of these methods and others, organizing them thematically to better highlight the current landscape and clarify the unique contributions of our proposed Fed-DIP framework. The revised section now provides a much stronger foundation for our paper and will include the following discussion:
>
>     The combination of large Vision-Language Models (VLMs) like CLIP [1] and parameter-efficient prompt learning [2] has advanced Federated Domain Generalization (FedDG). This approach reduces communication by having clients collaboratively train lightweight prompts instead of full models, but faces the challenge of aggregating knowledge from heterogeneous data. A common strategy is decomposing prompts into global and local components to balance generalization and personalization, using methods like specific learning objectives [3], Optimal Transport for alignment [4], or adaptive prompts [5]. More advanced architectures use Mixture of Experts (MoE) for personalization [6] or multi-modal visual prompts for richer representations [7]. A key distinction lies in prompt selection versus generation. While some generative methods use textual domain descriptions [8], our Fed-DIP uniquely synthesizes prompts directly from local visual features. This provides a more fine-grained, instance-adaptive guidance that captures nuanced domain variations, addressing a critical gap in prior work.

---

> ### Author Response · Authors · 2025-11-21
>
> [1] Radford, Alec, et al. "Learning transferable visual models from natural language supervision." International conference on machine learning. PmLR, 2021.
>     [2] Lester, Brian, Rami Al-Rfou, and Noah Constant. "The power of scale for parameter-efficient prompt tuning." arXiv preprint arXiv:2104.08691 (2021).
>     [3] Cui, Tianyu, et al. "Harmonizing generalization and personalization in federated prompt learning." arXiv preprint arXiv:2405.09771 (2024).
>     [4] Liu, Yifan, et al. "Global and local prompts cooperation via optimal transport for federated learning." Proceedings of the IEEE/CVF Conference on Computer Vision and Pattern Recognition. 2024.
>     [5] Fang, Chengying, et al. "FedPHA: Federated Prompt Learning for Heterogeneous Client Adaptation." Forty-second International Conference on Machine Learning. 2024.
>     [6] Luo, Jun, Chen Chen, and Shandong Wu. "Mixture of experts made personalized: Federated prompt learning for vision-language models." arXiv preprint arXiv:2410.10114 (2024).
>     [7] Singha, Mainak, et al. "FedMVP: Federated Multi-modal Visual Prompt Tuning for Vision-Language Models." arXiv preprint arXiv:2504.20860 (2025).
>     [8] Qiu, Chen, et al. "Federated text-driven prompt generation for vision-language models." The Twelfth International Conference on Learning Representations. 2024.
>
> 2.  **$\beta$ is missing in eq. 15**: Thank you for catching this typo. The correct total loss is indeed $L = L_d + \beta \cdot L_{cl}$. We will correct Equation 15 in the manuscript.
>
> ## Responses to Questions
>
> 1.  **Question:** How are the generated prompts through visual guided text adapter, integrated with class tokens, as the total input tokens in CLIP are limited?
>
>     **Our Response:**
>     The model integrates visual information into the text prompts and manages the fixed token limit of CLIP.
>
>    * **Visually-Guided Text Adaptation**:
>      *   During the `forward` pass, the model can receive `visual_features` from the image encoder.
>      *   These features are passed through a `visual_adapter`—a small neural network—which transforms the visual information into a format compatible with the text prompt embeddings.
>      *   The output of this adapter is then added to the learnable `ctx` vectors.  This fusion (`ctx = ctx + visual_features`) enriches the prompt with context from the input image, making it "visually-guided."
>
>    * **Integration with Class Tokens and Managing Token Limits**:
>      *   The integration happens at the embedding level.  The final prompt is an assembly of embeddings: the `[SOS]` token, the (now visually-guided) `ctx` vectors, and the `[CLASS_NAME] + [EOS]` tokens.
>      *   CLIP's constraint of a 77-token limit is handled during the initialization of the `PromptLearner`.
>      *   The number of learnable context tokens (`n_ctx`) is a fixed hyperparameter (e.g., 16).
>      *   When prompts are tokenized in the `__init__` method (`prompts = [prompt_prefix + " " + name + "." for name in classnames]`), the `clip.tokenize()` function automatically truncates any prompt that exceeds 77 tokens.
>      *   Therefore, the combined length of the context prefix and the class name must be chosen to fit within the 77-token limit.  If a class name is too long, it will be truncated, ensuring the input to CLIP's transformer never exceeds the maximum length.  The `token_suffix` buffer stores these potentially truncated class and EOS token embeddings.
>
>    By defining the number of learnable tokens (`n_ctx`) upfront and relying on CLIP's native tokenization and truncation, the architecture ensures that the total length of the assembled prompt `([SOS] + ctx + class_tokens + [EOS])` will always be within CLIP's operational limits.
>
> Please read "Algorithm 1" for specific details:

---

> ### Author Response · Authors · 2025-11-21
>
> ---
> **Algorithm 1: CAPE: Context-Aware Prompt Encoder**
>
> **Initialization**
>
> **Input:** A list of `classnames`, a pre-trained `clip_model`, number of context tokens `n_ctx`, an optional initial context string `ctx_init` (default: "a photo of a").
>
> - Define `n_cls` as the number of classes and `ctx_dim` as the CLIP model's feature dimension.
> - **Initialize Context Vectors (`ctx_vectors`):**
>     - Tokenize `ctx_init`.
>     - Use `clip_model` to get the token embeddings.
>     - Set `ctx_vectors` to the embeddings corresponding to the words in `ctx_init`.
>     - The `prompt_prefix` is `ctx_init`.
> - Make `ctx_vectors` a learnable parameter, referred to as `ctx`.
> - **Construct Full Prompts:**
>     - For each `classname`, create a prompt string: `prompt = prompt_prefix + " " + classname + "."`.
>     - Tokenize all these prompts and get their embeddings using `clip_model`.
> - From the full prompt embeddings, extract and register two non-learnable buffers:
>     - `token_prefix`: The embedding for the start-of-sequence (SOS) token.
>     - `token_suffix`: The embeddings for the class name and end-of-sequence (EOS) tokens.
> - **Initialize Visual Adapter:**
>     - Create a small neural network `visual_adapter` (Linear -> GELU -> Linear) to process visual features.
>
> **Forward Pass**
>
> **Input:** Optional `visual_features` from an image encoder.
>
> **Output:** A batch of prompt embeddings for the text encoder.
>
> - Start with the learnable context vectors `ctx`.
> - Expand `ctx` to have a batch dimension equal to `n_cls`.
> - **Fuse Visual Features:**
>     - Process `visual_features` through the `visual_adapter`.
>     - Add the resulting adapted features to the `ctx` vectors. This injects visual information into the text prompt context.
> - **Assemble Final Prompts:**
>     - Concatenate the `token_prefix`, the (potentially updated) `ctx`, and the `token_suffix` along the token dimension.
>     - `prompts = [token_prefix, ctx, token_suffix]`
> - Return the assembled `prompts`.
> ---
>
> 2.  **Question:** Why does MIDD consider multiple level of scales? Is the multi-scale distillation truly improves disentanglement compared to single-scale or does it overfit?
>
>     **Our Response:** The motivation for using a multi-scale approach in MIDD is to enable a more robust and fine-grained feature disentanglement. Visual features in images exist at different levels of abstraction and locality. For instance, high-level, global features might correspond to the object's core identity (e.g., "a car"), which should be domain-invariant. In contrast, low-level, local features might correspond to stylistic elements like texture, lighting, or artistic rendering (e.g., a "cartoon" car vs. a "photorealistic" car), which are often domain-specific.A single-scale distillation, especially one operating at a global level (i.e., on the entire feature map), may struggle to differentiate between these granularities. It might incorrectly penalize a local model for diverging on stylistic, domain-specific details that are genuinely unique to its data, or it might fail to enforce consensus on more subtle, region-specific invariant features. Regarding overfitting, we conduct experiments on multiple scales and find that its impact on the final accuracy rate is small.
>
> 3.  **Question:** What is the effect of $\lambda$ in eq. 7 ? Why is it not considered in the ablation study of section 5.2?
>
>     **Our Response:** The hyperparameter $\lambda$ in Eq. 7 is the weight for the total multi-scale distillation loss $L_{MKD}$. It controls the balance between the standard classification objective ($L_{CE}$) and our proposed knowledge disentanglement mechanism. We linearly ramp up $\lambda$ from 0 to 1 during training. This strategy ensures that the model first establishes stable training on the primary classification task before the distillation pressure is gradually applied to guide the feature separation. We did not include $\lambda$ in the hyperparameter analysis in Section 5.2 because its effect is part of a dynamic schedule rather than a fixed value. Its primary role is to ensure training stability rather than tuning the final model behavior, which is more directly influenced by parameters like $K$, $\tau$, and $\beta$.
>
>
> 4.  **Question:** Current works like FedOTP [1], FedMVP [2], FedPHA [3] should be considered as baselines.
>
>     **Our Response:** Thank you for this suggestion. These are indeed highly relevant and recent works. We agree that including them as baselines would provide a more comprehensive comparison. We will include the results in the final version of the paper.

---

### Official Review · Reviewer_oTKo · 2025-11-08

**Soundness:** 3
**Presentation:** 2
**Contribution:** 3
**Rating:** 6
**Confidence:** 4

**Summary:**

This paper introduces Fed-DIP (Federated Domain Generalization via Implicit Disentangled Distillation and Generative Prompting), a framework that enhances the generalization ability of vision-language models (VLMs) under federated non-IID settings. The approach has two main components: (1) Multi-scale Implicit Disentangled Distillation (MIDD), which separates global and local knowledge by aligning teacher-student logits at multiple spatial scales without exchanging raw data, and (2) Context-Aware Prompt Encoder (CAPE), which dynamically generates text-side prompts conditioned on instance-level visual context. Through selective gradient routing, the domain-invariant adapter $A_{di}$ is optimized toward shared global consensus, while the domain-specific adapter $A_{ds}$ captures local domain characteristics that are not aggregated. The proposed method is evaluated on PACS, VLCS, OfficeHome, and DomainNet, showing consistent improvements over existing federated domain generalization and prompt-based methods, including FedCLIP, PromptFL, and FedAPT. Ablation and sensitivity analyses further highlight the complementary roles of MIDD and CAPE and demonstrate favorable trade-offs between performance and communication efficiency.

**Strengths:**

- Conceptual novelty: The method unifies two important directions, federated domain generalization and prompt-based VLM adaptation, in a coherent framework.

- Algorithmic clarity in high level design: The MIDD component provides a clean way to separate global and local knowledge without data sharing, while CAPE dynamically conditions textual prompts on visual context.

- Strong empirical results: The model consistently outperforms both CNN-based and ViT-based federated DG methods on four benchmarks.

- Efficiency: Only lightweight adapters are communicated, achieving competitive results with significantly reduced communication cost.

**Weaknesses:**

- Missing formal definitions: The similarity weight $\gamma(s,i)$ is never formally defined. Its computation method must be clearly specified.
- Inconsistency in loss scheduling: The description of the loss weight $\lambda$ scheduling and the omission of the coefficient $\beta$ in the total loss equation create confusion about the actual implementation.
- Teacher mechanism ambiguity: The paper mixes two implementations (local teacher vs. server-broadcasted logits), leading to ambiguity about where and how teacher logits are computed.
- Communication cost mismatch: Reported values (2.019 MB vs 3.399 MB per round) conflict across text and figures, undermining reproducibility of the claimed efficiency.
- CAPE under-specified: The number of prompt tokens, their layer-wise injection strategy, and whether prompts are shared across classes are not detailed, which limits reproducibility.
- Notation and readability: Several key symbols are undefined at first use; a concise notation table would improve clarity.
- Related work coverage: The paper omits foundational prompt-learning works such as *CoOp* [1], *CoCoOp* [2], and *ProDA* [3], which are essential for positioning CAPE’s generative contribution.
- Privacy discussion missing: The logit exchange step introduces potential information leakage that is not acknowledged or mitigated. Discussion of possible privacy-preserving mechanisms would be valuable.

[1] Zhou, Kaiyang, et al. "Learning to prompt for vision-language models." International Journal of Computer Vision 130.9 (2022): 2337-2348.

[2] Zhou, Kaiyang, et al. "Conditional prompt learning for vision-language models." Proceedings of the IEEE/CVF conference on computer vision and pattern recognition. 2022.

[3] Lu, Yuning, et al. "Prompt distribution learning." Proceedings of the IEEE/CVF conference on computer vision and pattern recognition. 2022.

**Questions:**

1. How exactly is $\gamma(s,i)$ computed and normalized? Which distance metric is used, and is temperature scaling applied?

2. Does the complementary KL term risk divergence when teacher and student outputs are nearly orthogonal? What stabilizers (e.g., entropy regularization or clipping) are employed?

3. Is the teacher model computed locally from the previous global parameters, or are averaged logits broadcast by the server? Please provide a single authoritative description.

4. How are the text prompts injected into CLIP’s text transformer? Are they concatenated as prefixes or inserted between class tokens?

5. Can the authors clarify the discrepancy in communication cost numbers and provide the exact accounting of each component?

6. Was any privacy-preserving mechanism (e.g., differential privacy, quantization, or noise) applied to the transmitted logits?

Please refer to the Weaknesses section for the remaining questions.

---

> ### Author Response · Authors · 2025-11-21
>
> # Response to Reviewer oTKo
>
> Dear Reviewer oTKo,
>
> Thank you for your thorough and insightful review of our paper. Your detailed feedback is immensely valuable and has helped us identify several key areas for improvement. We appreciate the recognition of our work's conceptual novelty, algorithmic clarity, and strong empirical results.
>
> We will address all the weaknesses and questions you raised in the revised manuscript.
>
> ## Responses to Weaknesses
>
> 1.  **Missing formal definition of $\gamma(s,i)$**: We apologize for this omission. The similarity score $\gamma(s,i)$ is computed using the cosine similarity between the teacher and student logit vectors for a given region, followed by a temperature-scaled normalization. We will add the following formal definition to Section 3.2:
>
>     The similarity score $\gamma(s,i)$ is computed as the cosine similarity between the regional logit vectors of the teacher ($\rho_T(s,i)$) and the student ($\rho_S(s,i)$). To control the sensitivity of the weighting, we apply a temperature parameter $\tau_\gamma$:
>     $$
>     \gamma(s,i) = \frac{1}{2} \left(1 + \frac{\rho_T(s,i) \cdot \rho_S(s,i)}{|\rho_T(s,i)| |\rho_S(s,i)|} \right)
>     $$
>     This maps the similarity score to a range of [0,1], providing a normalized weight for the consistency and complementarity losses.
>
> 2.  **Inconsistency in loss scheduling and missing $\beta$**: We apologize for the confusion. The description of $\lambda$ scheduling was indeed inconsistent between the text and the implementation details. The implementation details are correct: $\lambda$ is linearly ramped up, and we will correct the text to reflect this. The omission of $\beta$ in the total loss equation was a typo. The correct total loss is $L = L_d + \beta \cdot L_{cl}$, which simplifies to $L = L_{CE} + \lambda \cdot L_{MKD} + \beta \cdot L_{cl}$. We will correct Equation 15 and add $\beta$ to our hyperparameter analysis in Section 5.2.
>
> 3.  **Teacher mechanism ambiguity**: We apologize for the ambiguity in the description of the Multi-scale Implicit Decoupling Distillation (MIDD) mechanism. To clarify the process:
>
>     - The server aggregates the domain-specific adapters of the clients to obtain the domain-invariant adapter of the server.
>     - The domain-specific adapters of the clients calculate the logit on the public dataset and pass it to the server.
>     - The server receives $logit_i$from multiple clients.
>     - The server calculates logit on the public dataset to obtain $logit_{server}$. Then, for each $logit_i$, multi-scale decoupling distillation is performed using $logit_server$and the public dataset label to update the model parameters on the server. During this process, client acts as the teacher and the server as the student for distillation.
>     - After the model parameters on the server are updated, the new $logit_{server}$is calculated on the public dataset and distributed to the client.
>     - On the client side, multi-scale decoupling distillation is performed using $logit_{client}$, $logit_{server}$, and the public dataset label to update the model parameters on the client side.During this process, server acts as the teacher and the client as the student for distillation.

---

> ### Author Response · Authors · 2025-11-21
>
> 4.  **Communication cost mismatch**:
>     The previous reported values (e.g, 2.019MB and 3.399MB) were incorrect. We will update the manuscript to reflect this corrected, two-part calculation to ensure clarity and reproducibility. We apologize for the confusion.
>
>     Communication overhead is generated in the following two processes:
>
>     **Transmission domain-specific adapter**
>
>     During communication, the client transmits its domain-specific adapter to the server, and then the server aggregates the domain-specific adapters received from various servers into domain-invariant adapters on the server.
>
>     *   Input/Output Dimension: 512
>     *   Number of Layers: 2
>     *   Bytes per Parameter: 4 (for a 32-bit float)
>     *   Number of Parameters per Layer: (Input Dimension $\times$ Output Dimension) + Bias Dimension = (512 $\times$ 512) + 512 = 262,656 parameters
>     *   Total Number of Parameters: Number of Layers $\times$ Parameters per Layer = 3 $\times$ 262,656 parameters = 787,968 parameters
>     *   Communication Cost (Bytes): Total Number of Parameters $\times$ Bytes per Parameter = 787,968 parameters $\times$ 4 bytes/parameter = 3,151,872 bytes
>     *   Communication Cost (MB): $\frac{3,151,872}{1024 \times 1024}$ $\approx$ 3.0058 MB
>
>     **Transmission logit**
>
>     After the aggregation operation is carried out, the server will perform the distillation operation. During this process, logit will be transmitted (for the detailed process, please refer to the previous "Weakness: Teacher mechanism ambiguity"). The communication overhead calculation process in this procedure is as follows:
>
>     **Calculation Formulas**
>
>     *   Batch Size: 128
>     *   Feature Dimension: 512
>     *   Bytes per Element: 4 (for a 32-bit float)
>     *   Total Number of Elements: Batch Size $\times$ Feature Dimension = 128 $\times$ 512 = 65,536 elements
>     *   Communication Cost (Bytes): Total Number of Elements $\times$ Bytes per Element = 65,536 elements $\times$ 4 bytes/element = 262,144 bytes
>     *   Communication Cost (MB): $\frac{262,144}{1024 \times 1024}$ = 0.25 MB
>
>     **Total Communication Cost(MB)**: The total communication overhead is the sum of the overhead of the specific adapter and logit in the transmission domain, which amounts to 3.0058 MB + 0.25 MB=3.2558 MB
>
> 5.  **CAPE under-specified**: We appreciate the reviewer's feedback on the need for greater clarity. This process is detailed in the following "Algorithm 1", which outlines the structure of our CAPE.
>
> ---
> **Algorithm 1: CAPE: Context-Aware Prompt Encoder**
>
> **Initialization**
>
> **Input:** A list of `classnames`, a pre-trained `clip_model`, number of context tokens `n_ctx`, an optional initial context string `ctx_init` (default: "a photo of a").
>
> - Define `n_cls` as the number of classes and `ctx_dim` as the CLIP model's feature dimension.
> - **Initialize Context Vectors (`ctx_vectors`):**
>     - Tokenize `ctx_init`.
>     - Use `clip_model` to get the token embeddings.
>     - Set `ctx_vectors` to the embeddings corresponding to the words in `ctx_init`.
>     - The `prompt_prefix` is `ctx_init`.
> - Make `ctx_vectors` a learnable parameter, referred to as `ctx`.
> - **Construct Full Prompts:**
>     - For each `classname`, create a prompt string: `prompt = prompt_prefix + " " + classname + "."`.
>     - Tokenize all these prompts and get their embeddings using `clip_model`.
> - From the full prompt embeddings, extract and register two non-learnable buffers:
>     - `token_prefix`: The embedding for the start-of-sequence (SOS) token.
>     - `token_suffix`: The embeddings for the class name and end-of-sequence (EOS) tokens.
> - **Initialize Visual Adapter:**
>     - Create a small neural network `visual_adapter` (Linear -> GELU -> Linear) to process visual features.
>
> **Forward Pass**
>
> **Input:** Optional `visual_features` from an image encoder.
>
> **Output:** A batch of prompt embeddings for the text encoder.
>
> - Start with the learnable context vectors `ctx`.
> - Expand `ctx` to have a batch dimension equal to `n_cls`.
> - **Fuse Visual Features:**
>     - Process `visual_features` through the `visual_adapter`.
>     - Add the resulting adapted features to the `ctx` vectors. This injects visual information into the text prompt context.
> - **Assemble Final Prompts:**
>     - Concatenate the `token_prefix`, the (potentially updated) `ctx`, and the `token_suffix` along the token dimension.
>     - `prompts = [token_prefix, ctx, token_suffix]`
> - Return the assembled `prompts`.
> ---
>
> 6.  **Notation and readability**: We agree that the paper's clarity can be improved. We will carefully review the paper.

---

> ### Author Response · Authors · 2025-11-21
>
> 7.  **Related work coverage**: Thank you for pointing out this important omission. You are absolutely right that a discussion of foundational works like CoOp, CoCoOp, and ProDA is essential for properly contextualizing the contribution of our CAPE module. We apologize for this oversight and will significantly revise our related work section to include a thorough discussion of these methods.
>
>     Specifically, we will add the following analysis to Section 2 (Related Work):
>
>     Prompt learning has revolutionized adapting large VLMs like CLIP [1]. Early work focused on the language side, with CoOp [2] learning continuous prompt vectors. CoCoOp [3] improved this by making prompts instance-conditional. In parallel, Visual Prompt Tuning (VPT) [4] introduced learnable tokens to the vision encoder. Hybrid approaches like MaPLe [5] prompted both encoders. To enhance robustness, ProDA [6] learned a prompt distribution for diverse sampling. Our CAPE module builds on these principles by generating instance-aware prompts conditioned on disentangled visual features, representing a novel approach in the federated context.
>
>     [1] Radford, Alec, et al. "Learning transferable visual models from natural language supervision." International conference on machine learning. PmLR, 2021.
>
>     [2] Zhou, Kaiyang, et al. "Learning to prompt for vision-language models." International Journal of Computer Vision 130.9 (2022): 2337-2348.
>
>     [3] Zhou, Kaiyang, et al. "Conditional prompt learning for vision-language models." Proceedings of the IEEE/CVF conference on computer vision and pattern recognition. 2022.
>
>     [4] Jia, Menglin, et al. "Visual prompt tuning." European conference on computer vision. Cham: Springer Nature Switzerland, 2022.
>
>     [5] Khattak, Muhammad Uzair, et al. "Maple: Multi-modal prompt learning." Proceedings of the IEEE/CVF conference on computer vision and pattern recognition. 2023.
>
>     [6] Lu, Yuning, et al. "Prompt distribution learning." Proceedings of the IEEE/CVF conference on computer vision and pattern recognition. 2022.
>
> 8.  **Privacy discussion missing**: We thank the reviewer for raising this important point. We selected logits sharing because it is a proven, privacy-preserving method that is safer than alternatives like sharing gradients or model parameters. This choice is a deliberate design decision to enhance the security of our framework. We will ensure this justification is clear in the revised manuscript.
>
>     The privacy advantages of this approach are documented in the literature, primarily due to its defense against several common attack vectors:
>
>     Defense against Reconstruction Attacks: Logits, as final prediction probabilities, are a high-level abstraction of data. Unlike gradients, they do not contain direct structural information about the input samples, making it difficult for an attacker to reconstruct the original private data [1].
>
>     Reduced Inference Risk: Sharing model parameters makes the system vulnerable to white-box attacks, such as membership inference. In contrast, sharing only the output distributions is more robust against such attacks, as the internal model state remains private [2].
>
>     Reduced Information Leakage: The dimensionality of logits is equal to the number of classes, which is smaller than the number of model parameters. This reduces the information exposed during communication, limiting the potential for information leakage and mitigating risks like data poisoning [2].
>
>     [1] Chang, Hong et al. “Cronus: Robust and Heterogeneous Collaborative Learning with Black-Box Knowledge Transfer.” ArXiv abs/1912.11279 (2019): n. pag.
>
>     [2] Li, Lin et al. “Federated Distillation: A Survey.” ArXiv abs/2404.08564 (2024): n. pag.

---

> ### Author Response · Authors · 2025-11-21
>
> ## Responses to Questions
>
> 1.  **Question:** How exactly is $\gamma(s,i)$ computed and normalized? Which distance metric is used, and is temperature scaling applied?
>
>     **Our Response:** As mentioned above, it is the cosine similarity between the regional teacher and student (domain-invariant branch) logit vectors, mapped to a range of [0, 1]. We do not use temperature scaling for this specific weight computation.
>
> 2.  **Question:** Does the complementary KL term risk divergence when teacher and student outputs are nearly orthogonal? What stabilizers (e.g., entropy regularization or clipping) are employed?
>
>     **Our Response:** This is an excellent technical question regarding the stability of the complementarity loss, $D_{comp} = -D_{\text{KL}}(\sigma(Z_{T}) \| \sigma(f_{ds}(z)))$. The concern is valid: maximizing the KL divergence could theoretically lead to unstable training or divergence if not properly managed.
>
>     Our framework incorporates several implicit and explicit stabilization mechanisms to mitigate this risk:
>
>     - **Exponential Ramping of $\lambda$:** As mentioned in Section 3.2, we ramp up the weight of the total distillation loss, $\lambda$, exponentially from a small initial value. This is a critical stabilizer. In the early stages of training, the model prioritizes the standard classification loss ($L_{CE}$). This ensures that the domain-specific adapter ($A_{ds}$) first learns to produce meaningful, task-relevant predictions before the complementarity pressure is strongly applied. This prevents the model from diverging at initialization when its outputs are likely to be noisy and far from the teacher's.
>     - **Shared Backbone and Initialization:** The domain-specific branch is not an entirely separate network. It consists of a lightweight adapter ($A_{ds}$) attached to a shared, pre-trained backbone. The features `z` that are input to the adapter are already semantically rich. This shared structure constrains the output space of the domain-specific branch, making it highly unlikely for its predictions to become truly orthogonal to the teacher's.
>     - **Bounded Output of Softmax:** The use of the softmax function $\sigma$ in the KL divergence calculation ensures that the probability distributions are always bounded within $[0, 1]$ and sum to 1. While this doesn't prevent large divergence values, it constrains the optimization landscape and prevents unbounded outputs.
>
>     In practice, we found that these inherent and designed stabilizers were sufficient to ensure robust and convergent training without needing additional explicit techniques like output clipping or entropy regularization. The combination of a gradual increase in distillation pressure and the architectural constraints of the shared backbone effectively prevents the complementarity loss from causing divergence.
>
> 3.  **Question:** Is the teacher model computed locally from the previous global parameters, or are averaged logits broadcast by the server? Please provide a single authoritative description.
>
>     **Our Response:** Thank you for pointing out this question. Please refer to "Weaknesses: Teacher mechanism ambiguity" above.
>
> 4.  **Question:** How are the text prompts injected into CLIP’s text transformer? Are they concatenated as prefixes or inserted between class tokens?
>
>     **Our Response:** This explanation details how text prompts are structured and injected into CLIP's text transformer, and how visually-guided prompts are integrated while respecting CLIP's token limit.
>
>     The `CAPE` module constructs prompts at the embedding level, rather than by manipulating text. This allows for end-to-end optimization. The process is as follows:
>
>    * **Initialization of Context Vectors (`ctx`)**:
>        *   A set of learnable vectors, referred to as `ctx`, is initialized. These vectors act as a "prompt" that is optimized during training to be most effective for the task.
>        *   The initialization can be done in two ways:
>            *   **Text-based (`ctx_init`)**: If a string like `"a photo of a"` is provided, it is tokenized, and its corresponding word embeddings from CLIP's vocabulary are used as the starting point for `ctx`.
>            *   **Random**: If no initial text is provided, `ctx` is initialized with random values.
>        *   These `ctx` vectors are registered as a `nn.Parameter`, making them trainable.
>
>      * **Static Prompt Components**:
>          *   For each class name (e.g., "dog"), a full prompt string is conceptualized as `[SOS] [CTX_VECTORS] [CLASS_NAME] [EOS]`.
>          *   The `[SOS]` (Start-of-Sequence) token embedding is stored in `self.token_prefix`.
>          *   The embeddings for the class name tokens (e.g., "dog", ".") and the `[EOS]` (End-of-Sequence) token are stored together in `self.token_suffix`. These are pre-computed and fixed for each class.

---

> ### Author Response · Authors · 2025-11-21
>
> * **Dynamic Prompt Assembly in `forward()`**:
>          *   In the `forward` pass, the final prompt embedding is assembled by concatenating the static and learned components in the correct order: `[prefix, ctx, suffix]`.
>          *   The resulting tensor has the shape `[number_of_classes, 77, embedding_dimension]`, which is the exact format expected by CLIP's text transformer. `77` is CLIP's maximum token sequence length.
>
>      This method bypasses text tokenization during the forward pass and directly constructs the input for the text transformer, allowing the `ctx` prompt to be learned through backpropagation.
>
>      Please read "Weaknesses: CAPE under-specified" for more information. The "CAPE" in Figure 2 of the paper describes this process.
>
> 5.  **Question:** Can the authors clarify the discrepancy in communication cost numbers and provide the exact accounting of each component?
>
>     **Our Response:** We apologize for the error. The previous reported values (e.g, 2.019MB and 3.399MB) both are incorrect. Please read "Weaknesses: Communication cost mismatch" for more information.
>
> 6.  **Question:** Was any privacy-preserving mechanism (e.g., differential privacy, quantization, or noise) applied to the transmitted logits?
>
>     **Our Response:** We chose logit sharing as an inherently privacy-preserving technique compared to gradient or parameter sharing, as it exposes less direct information about the source data. In the current implementation, we did not apply additional mechanisms like differential privacy or noise, but we acknowledge this as a valuable direction for future work to further enhance security guarantees.Please read "Weaknesses: Privacy discussion missing" for more information.

---

### Meta-Review · Area_Chair_5Eo5 · 2025-12-31

**Summary:**

The paper introduces a federated learning framework FedDIP for domain generalization, which integrates two key modules: Multi-scale Implicit Decoupling Distillation (MIDD) and Context-Aware Prompt Encoder (CAPE). MIDD aligns intermediate features across clients to promote domain-invariant representation learning, and CAPE leverages prompt-based conditioning to adapt local models to unseen domains.

The quality of the submissions needs significant improvement. Multiple reviewers point out that the original submission contained numerous typos or omissions of important elements. Some key mechanisms (e.g., teacher mechanism) are unclear or ambiguous. Some important experimental values (e.g., communication cost) are mismatched or incorrect. Several critical parameters or settings (e.g., the number of prompt tokens, their layer-wise injection strategy) are not clearly stated, which limits reproducibility. Important or foundational related work are not introduced. Although some issues can be corrected in a revised version, they reflect that this submission may not have been carefully written and adequately checked.

Additionally, the author did not explicitly provide evidence to directly demonstrate whether multi-scale distillation truly improves disentanglement compared to single-scale distillation. The author did not provide clear support to show that the linear increment strategy for $\lambda$ is optimal (compared to other possible strategies). Recent works (e.g., FedOTP, FedMVP, and FedPHA) are not utilized as baselines. The original submission lacks a quantitative analysis of whether the extracted features actually exhibit domain invariance or reduced domain discrepancy. The added comparative experiment lacks statistical analysis to determine whether the gains exhibit statistical significance. and the ablation experiments may be not complete.

Based on the above considerations, I think this manuscript does not match the ICLR’s requirement and I do not recommend to accept this manuscript.

**Reviewer Concerns:**

The concerns raised by Reviewer PwF6 may have not been adequately addressed. The author did not explicitly provide evidence to directly demonstrate whether multi-scale distillation truly improves disentanglement compared to single-scale distillation. The author did not provide clear support to show that the linear increment strategy for $\lambda$ is optimal (compared to other possible strategies). Recent works (e.g., FedOTP, FedMVP, and FedPHA) are not utilized as baselines. The concerns raised by Reviewer oWjg may have not been adequately addressed. The added comparative experiment lacks statistical analysis to determine whether the gains exhibit statistical significance.

**Reviewer Scores:**

Reviewers would keep their scores.

---

### Decision · Program_Chairs · 2026-01-26

Reject